# Reinforce LLM Reasoning through Multi-Agent Reflection

**Yurun Yuan**[1]   **Tengyang Xie**[1]

## Abstract

Leveraging more test-time computation has proven to be an effective way to boost the reasoning capabilities of large language models (LLMs). Among various methods, the verify-and-improve paradigm stands out for enabling dynamic solution exploration and feedback incorporation. However, existing approaches often suffer from restricted feedback spaces and lack of coordinated training of different parties, leading to suboptimal performance. To address this, we model this multi-turn refinement process as a Markov Decision Process and introduce DPSDP (**D**irect **P**olicy **S**earch by **D**ynamic **P**rogramming), a reinforcement learning algorithm that trains an actor-critic LLM system to iteratively refine answers via direct preference learning on self-generated data. Theoretically, DPSDP can match the performance of any policy within the training distribution. Empirically, we instantiate DPSDP with various base models and show improvements on both in- and out-of-distribution benchmarks. For example, on benchmark MATH 500, majority voting over five refinement steps increases first-turn accuracy from 58.2% to 63.2% with Ministral-based models. An ablation study further confirms the benefits of multi-agent collaboration and out-of-distribution generalization.

## 1. Introduction

Large language models (LLMs) have shown strong capabilities in solving reasoning tasks such as mathematical problems and coding (Lozhkov et al., 2024; Shao et al., 2024; Team et al., 2024). A series of effective methods to improve the reasoning performance of LLMs involve leveraging additional computation at inference time. Mechanisms such as best-of-N sampling (Charniak and Johnson, 2005; Stiennon et al., 2020; Chow et al., 2024), self-consistency (Wang et al., 2022), explicit reasoning (Wei et al., 2022; Yao et al., 2023; OpenAI, 2024), and refinement through critique and revision (Qu et al., 2024; Kumar et al., 2024) exemplify this approach.

Among the various test-time scaling methods, the verify-and-improve paradigm offers a distinct advantage by enabling interaction with external environments to explore the solution space and incorporate feedback into response generation. For instance, when faced with tasks like using an unfamiliar coding library that is not covered in the LLM agent's knowledge, the LLM agent needs to compose trial programs and interact with the compiler and tests to refine its attempts. In contrast, methods like chain-of-thought and self-consistency remain confined to the knowledge already embedded within the LLM, limiting their adaptability in dynamically evolving tasks.

Various prior works have explored methods to enhance LLMs' ability to refine their responses. One line of research focuses on enabling LLMs to correct their own errors by extracting and utilizing latent knowledge that was not effectively applied in the initial attempt (Kumar et al., 2024; Qu et al., 2024), despite the findings from other works that LLMs is unable to reliably correct their errors without external information (Kamoi et al., 2024; Huang et al., 2024). Another line of research introduces external feedback mechanisms to guide refinement, leveraging resources such as compilers, external tools, and verifier models (Welleck et al., 2022; Havrilla et al., 2024; Chen et al., 2024a; Shinn et al., 2023). By incorporating external feedback, LLMs gain access to new information that can improve their prior responses. Nonetheless, these approaches face limitations such as: **(1)** a restricted feedback space (e.g., compiler messages in code generation tasks or output of a fixed set of tools for tool-assisted refinement), and **(2)** the absence of joint training processes among LLM agents and feedback providers, resulting in suboptimal interaction between LLMs and their environments.

Recent progress in multi-agent systems have shown great promise across various tasks, highlighting new opportunities for response improvement (Guo et al., 2024b; Mot-

---

[1]Department of Computer Sciences, University of Wisconsin–Madison, Madison, WI, USA. Correspondence to: Yurun Yuan <yurun_yuan@cs.wisc.edu>, Tengyang Xie <tx@cs.wisc.edu>.

*Proceedings of the 42$^{nd}$ International Conference on Machine Learning*, Vancouver, Canada. PMLR 267, 2025. Copyright 2025 by the author(s).

wani et al., 2024). Instead of treating feedback resources as fixed, multi-agent systems incorporate different parties into the training-time optimization process, enabling better coordination among agents. Building on this foundation, we propose DPSDP: **D**irect **P**olicy **S**earch by **D**ynamic **P**rogramming, an RL algorithm designed to train a multi-agent LLM system to iteratively improve its responses for reasoning tasks with self-generated data. DPSDP introduces an actor model that generates and refines responses over multiple turns, guided by feedback from a critic model at each turn. This approach enables a broad and flexible feedback space by utilizing the diverse and dynamic responses generated by LLM agents. Furthermore, the joint training process optimizes the collaboration between the actor and critic, harnessing the strengths of both models to achieve more effective response refinement.

**Contribution**   Our contributions are three-fold. First, we introduce DPSDP, an RL algorithm that enables LLMs to iteratively refine responses with collaboration. We formulate the multi-turn improvement process as a Markov Decision Process (MDP) and design a direct preference learning algorithm to teach LLMs from self-generated data. Furthermore, we theoretically prove that the policy produced by DPSDP competes with any policy under single-policy concentrability and bounded in-distribution generalization error. Finally, we demonstrate the effectiveness of our algorithm by instantiating our method across various model families, including Ministral (Mistral AI team, 2024), Llama-3.1 (Grattafiori et al., 2024), and Qwen2.5 (Yang et al., 2024a), and evaluating their performance on multiple benchmarks. Specifically, by sampling five sequential answers on problems from MATH 500, Ministral-based models improve their first-turn accuracy from 58.2% to 63.2%, Llama-3.1-based models from 55.8% to 58.4%, and Qwen2.5-based models from 60.4% to 62.0%. We also show that our models generalize effectively to out-of-distribution benchmarks, from grade-school-level problems to challenging Olympiad-level benchmarks.

Additionally, we explore the factors that contribute to the strong empirical performance. First, we replicate the training process on a single LLM agent, and our findings indicate that while training a single LLM to both solve questions and reflect on them can improve reasoning capabilities, it struggles with more challenging benchmarks, such as MATH 500 and the Olympiad Bench, highlighting the benefits of specialized LLM agents. Furthermore, we evaluate the models in non-Markovian settings, where agents have access to the full history of prior refinement iterations. This setup deviates from the MDP defined by our framework, which assumes states are based only on the most recent answer. While providing the full conversation history offers richer context, our results show that this setting

induces greater distribution shift and leads to degraded performance.

## 2. Problem Setup and Preliminaries

Consider a scenario involving two agents: an actor $\pi_a$, which generates answers to questions, and a critic $\pi_c$, which provides feedback on the actor's responses to help refine them. Let $\mathcal{D}_{\text{prob}} = \{(\boldsymbol{x}_i, \boldsymbol{a}_i^*)\}_{i=1}^N$ represent a dataset of $N$ questions and corresponding ground truth answers, with $\boldsymbol{x}, \boldsymbol{a}^* \sim \mathcal{D}_{\text{prob}}$ denoting a question posed by a human user along with its correct answer. When the actor receives the question $\boldsymbol{x}$, it generates an initial response $a_0 \sim \pi_a(\cdot \mid \boldsymbol{x})$. The critic then evaluates this response, providing feedback $a_1 \sim \pi_c(\cdot \mid [\boldsymbol{x}, a_0])$ that identifies potential errors or inaccuracies. Using this feedback, the actor refines its response, generating $a_2 \sim \pi_a(\cdot \mid [\boldsymbol{x}, a_0, a_1])$. This critique-and-refine process can be repeated multiple times, enabling the actor to iteratively improve its answers. The final output is determined by majority voting across all answers. We illustrate this process in Figure 1.

**Cast to a Markov Decision Process (MDP)**   This multi-turn conversation can be naturally formulated as a standard episodic Markov Decision Process (MDP). Let $\pi = (\pi_a, \pi_c)$ represent the joint policy governing the iterative refinement process. The state at turn $h$, denoted as $s_h$, captures the conversation history up to that point. The actions in this MDP correspond to the responses $a_h$ generated by either the actor or the critic at each turn, and the policy $\pi$ determines the action taken given the current state, i.e., $a_h \sim \pi(\cdot \mid s_h)$. The new state $s_{h+1}$ incorporates the new response $a_h$ and full or partial conversation history from last state. We use a deterministic function $\delta(s_h, a_h)$ to integrate the new response into the conversation history to produce the next state: $s_{h+1} = \delta(s_h, a_h)$. We define the state space at turn $h$ as $\mathcal{S}_h$ and action space as $\mathcal{A}_h$. The state distribution over $\mathcal{S}_h$ induced by policy $\pi$ is represented as $d_h^\pi$. If the critique-and-refine process is performed $L$ times, the MDP will have a horizon of $H = 2L + 1$, comprised of an initial answer from the actor and following $L$ rounds of feedback from the critic and refinements by the actor. The reward function $r(s)$ incentivizes the actor to produce correct answers. Specifically, a reward is assigned when the actor's response matches the ground truth answer $\boldsymbol{a}^*$, defined as $r(s_{2i+1}) = \mathbb{I}[a_{2i} = \boldsymbol{a}^*]$ for $i \in \{0, 1, \cdots, L\}$.

We also define the value functions $Q_h^\pi(s, a) = \mathbb{E}_\pi\big[\sum_{t=h}^{H-1} r(s_t) \mid s_h = s, a_h = a\big]$ and $V_h^\pi(s) = \mathbb{E}_{a \sim \pi(\cdot \mid s)}\big[Q_h^\pi(s, a)\big]$ as the expected undiscounted cumulative returns starting at step $h$ for $h \in [H]$. Specifically, we define $\mathcal{J}(\pi)$ as the expected return of the entire trajectory and we aim to learn a policy that maximize the number of correct answers: $\max_\pi \mathcal{J}(\pi) := \mathbb{E}_{s_0 \sim d_0^\pi}\big[V_0^\pi(s)\big]$, where

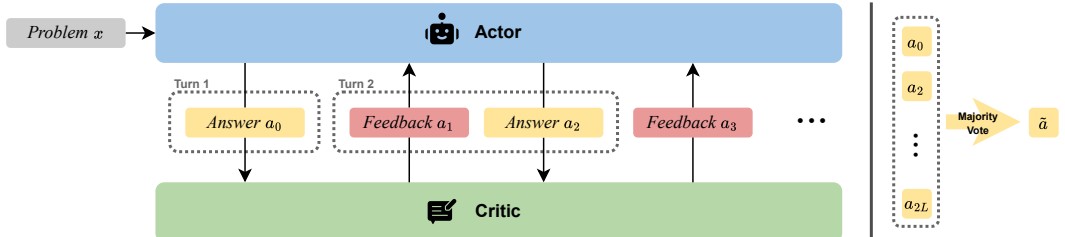

*Figure 1.* Inference time. Given a problem $x$, the actor $\pi_a$ generates an initial response $a_0$. The critic $\pi_c$ then provides feedback $a_1$, identifying potential errors in $a_0$. The actor iteratively refines its response based on the feedback, continuing this process for $L$ rounds. Finally, majority voting is applied to all generated answers to determine the final response $\tilde{a}$.

$d_0^\pi$ denotes the distribution over initial states.

**Policy Search by Dynamic Programming (PSDP)** PSDP (Bagnell et al., 2003) is a classic reinforcement learning algorithm designed to optimize undiscounted rewards over a fixed horizon. For a non-stationary policy $\pi_{\mathsf{NS}} = \left(\pi^0, \pi^1, \ldots, \pi^{H-1}\right)$, where each $\pi^h \in \Pi$ is a stationary policy deployed at time step $h$, the PSDP algorithm shown as in Algorithm 3 iteratively optimizes policies in reverse order from $h = H - 1$ to $h = 0$, selecting $\pi^h$ from $\Pi$ at each step to maximize the expected future rewards following the policy sequence $(\pi^h, \pi^{h+1}, \ldots, \pi^{H-1})$, starting with states sampled from a baseline distribution $\mu_h$. PSDP provides a theoretical performance guarantee, even when the maximization step $\arg\max_{\pi \in \Pi}$ can be done only approximately (Bagnell et al., 2003).

## 3. DPSDP: Direct Policy Search by Dynamic Programming

In this section, we develop DPSDP, an RL algorithm inspired by PSDP, to enable LLMs to generate enhanced answers via collaboration. We first induce an ideal version of our algorithm, consisting of data collection and $Q$-value learning. We then provide a theoretical proof for performance guarantees. Lastly, we modify the algorithm to obtain a practical version which accommodates implementation difficulty and computational efficiency.

### 3.1. Algorithm Development

We start with a reference policy $\pi_{\mathsf{ref}}$, which represents LLM agents that are capable of but not specialized in reasoning or response refinement.

Motivated by PSDP, we want to maximize the expected return at each turn following the optimized policy from the last iteration, i.e., $\pi_h(\cdot \mid s_h) = \arg\max_{a \in \mathcal{A}_h} Q_h^{\pi_{h+1}}(s_h, a)$. However, accurately solving this requires a finite action space, which is unrealistic for a model's open-ended responses. We alternatively consider a

KL-regularized objective with respect to $\pi_{\mathsf{ref}}$. Specifically, for each fixed $h \in \{H-1, H-2, \cdots, 0\}$, we aim to obtain a policy $\widehat{\pi}_h$ such that

$$
\begin{aligned}
\widehat{\pi}_h = \max_{\pi \in \Delta(\mathcal{A}_h)} \; & \mathbb{E}_{s_h \sim d_h^{\pi_{\mathsf{ref}}}, a_h \sim \pi(\cdot|s_h)} \big[ Q_h^{\widehat{\pi}_{h+1}}(s_h, a_h) \\
& - \beta D_{\mathrm{KL}}[\pi(\cdot \mid s_h) \| \pi_{\mathsf{ref}}(\cdot \mid s_h)] \big].
\end{aligned}
\tag{1}
$$

Here, $D_{\mathrm{KL}}$ denotes the Kullback-Leibler divergence and parameter $\beta$ is used to balance the gap to the unregularized objective. Prior works show that there exists a closed-from solution to this problem: $\widehat{\pi}_h(a \mid s_h) \propto \pi_{\mathsf{ref}}(a \mid s_h) \exp\left(\frac{1}{\beta} Q_h^{\widehat{\pi}_{h+1}}(s_h, a)\right)$, and this objective can be learned in a direct learning manner (Rafailov et al., 2023; Rosset et al., 2024).

Particularly, we define the cross-entropy loss as

$$
\begin{aligned}
& \mathcal{L}_{\mathsf{CE}}\left(\pi, \pi_{\mathsf{ref}}; \rho, Q\right) \\
= & \mathbb{E}_{(s, a^1, a^2) \sim \rho} \Bigg\{ H_{\mathbb{B}}\bigg( \sigma\left(Q(s, a^1) - Q(s, a^2)\right), \\
& \sigma\left( \beta \log \frac{\pi(a^1 \mid s)}{\pi_{\mathsf{ref}}(a^1 \mid s)} - \beta \log \frac{\pi(a^2 \mid s)}{\pi_{\mathsf{ref}}(a^2 \mid s)} \right) \bigg) \Bigg\},
\end{aligned}
\tag{2}
$$

where $H_{\mathbb{B}}(z, \hat{z}) := -z \log \hat{z} - (1-z) \log(1-\hat{z})$ is the cross entropy of two Bernoulli distributions. With a dataset $\mathcal{D}_h$ consisting of $\left(s_h, a_h^1, a_h^2\right)$, where $s_h \sim d_h^{\pi_{\mathsf{ref}}}$, $a_h^i \sim \pi_{\mathsf{ref}}(\cdot \mid s_h)$, $i \in \{1, 2\}$, we can achieve Eq. (1) via optimizing $\min_\pi \mathcal{L}_{\mathsf{CE}}\left(\pi, \pi_{\mathsf{ref}}; \mathbb{U}[\mathcal{D}_h], Q_h^{\widehat{\pi}_{h+1}}\right)$. In brief, at each step, DPSDP first samples various responses from $\pi_{\mathsf{ref}}$, then obtains a policy that minimizes the cross-entropy loss Eq. (2) over the collected pairwise dataset. The reference policy $\pi_{\mathsf{ref}}$ induces the baseline distributions. We formally outline DPSDP algorithm in Algorithm 1.

### 3.2. Algorithm Analysis

In this section, we analyze the performance of DPSDP under appropriate assumptions. We first assume that $\pi_{\mathsf{ref}}$ provides fair coverage of the state distribution of the optimal

**Algorithm 1** DPSDP

**Input:** horizon $H$, reference policy $\pi_{\mathsf{ref}}$, dataset $\mathcal{D}_{\mathsf{prob}}$, and $Q$-value functions $Q_h^\pi(s, a)$

 **for** $h = H - 1, H - 2, \ldots, 0$ **do**
  Initialize $\mathcal{D}_h \leftarrow \emptyset$.
  **for** $(\boldsymbol{x}, \boldsymbol{a}^*) \in \mathcal{D}_{\mathsf{prob}}$ **do**
   Sample $m$ pairs $\left\{ \left( s_h, a_h^1, a_h^2 \right) \right\}_m$, where $s_h \sim d_h^{\pi_{\mathsf{ref}}}(\cdot \mid s_0 = \boldsymbol{x})$, $a_h^1 \sim \pi_{\mathsf{ref}}(\cdot \mid s_h)$, $a_h^2 \sim \pi_{\mathsf{ref}}(\cdot \mid s_h)$.
   Collect the pair: $\mathcal{D}_h \leftarrow \mathcal{D}_h \cup \left\{ \left( s_h, a_h^1, a_h^2 \right) \right\}_m$.
  **end for**
  Let $\widehat{\pi}_h \leftarrow \min_\pi \mathcal{L}_{\mathsf{CE}} \left( \pi, \pi_{\mathsf{ref}}; \mathbb{U}[\mathcal{D}_h], Q_h^{\widehat{\pi}_{h+1}} \right)$
  where $\mathcal{L}_{\mathsf{CE}}$ is defined as Eq. (2).
 **end for**
 Let the final policy be $\widehat{\pi} = \widehat{\pi}_0$.

**Ouput:** DPSDP policy $\widehat{\pi}$

policy $\pi^\star$ and generates more diverse responses than other policies, as formalized in Assumption 1.

**Assumption 1** (Coverage). *We assume the concentrability coefficients with respect to space distribution and action space defined below are bounded, i.e., $C_\mathcal{S}^\star < +\infty$ and $C_\mathcal{A} < +\infty$.*

$$C_\mathcal{S}^\star := \max_{h, s_h} \frac{d_h^{\pi^\star}(s_h)}{d_h^{\pi_{\mathsf{ref}}}(s_h)}, \quad C_\mathcal{A} := \max_{\pi \in \Pi} \max_{h, s_h} \frac{\pi(a_h \mid s_h)}{\pi_{\mathsf{ref}}(a_h \mid s_h)}.$$

This ensures the training data generated from $\pi_{\mathsf{ref}}$ is sufficiently exploratory, providing an opportunity for DPSDP to learn the optimal actions.

Additionally, we assume that Algorithm 1 maintains a bounded in-distribution generalization error, as stated in Assumption 2.

**Assumption 2** (In-distribution reward learning). *We assume the policy $\widehat{\pi}$ obtained with Algorithm 1 satisfies that for any $h \in \{0, 1, 2, \cdots, H - 1\}$,*

$$\mathbb{E}_{s_h \sim d_h^{\pi_{\mathsf{ref}}}, a_h \sim \pi_{\mathsf{ref}}(\cdot|s_h), a_h' \sim \pi_{\mathsf{ref}}(\cdot|s_h)} \left[ \left( \beta \log \frac{\widehat{\pi}(a_h \mid s_h)}{\pi_{\mathsf{ref}}(a_h \mid s_h)} - \right. \right.$$

$$\left. \left. \beta \log \frac{\widehat{\pi}(a_h' \mid s_h)}{\pi_{\mathsf{ref}}(a_h' \mid s_h)} - Q_h^{\widehat{\pi}}(s_h, a_h) + Q_h^{\widehat{\pi}}(s_h, a_h') \right)^2 \right] \leq \varepsilon_{\mathsf{stat}}.$$

By Lemme C.5 of Xie et al. (2024), we can immediately infer that $\varepsilon_{\mathsf{stat}}$ is small with high probability for large dataset.

Under these assumptions, we establish the following performance guarantees for DPSDP.

**Theorem 1.** *Under Assumptions 1 and 2, if we choose $\beta = O\left( \frac{\sqrt{C_\mathcal{S}^\star C_\mathcal{A} \varepsilon_{\mathsf{stat}}}}{\log C_\mathcal{A}} \right)$, then DPSDP policy $\widehat{\pi}$ satisfies $\mathcal{J}(\pi^\star) - \mathcal{J}(\widehat{\pi}) = O\left( H \sqrt{C_\mathcal{S}^\star C_\mathcal{A} \varepsilon_{\mathsf{stat}}} \right).$*

The policy produced by DPSDP theoretically competes with any policy under single-policy concentrability and bounded in-distribution generalization error. We defer the proof of Theorem 1 to Appendix C.

### 3.3. Practical Algorithm

While Algorithm 1 achieves a theoretical performance guarantee, we also aim for a practical and efficient implementation. Achieving this requires carefully refining several key aspects.

**Reduced Context and Horizon Generalization** Previous works that model multi-turn conversations as MDPs maintain the full conversation history at each state, i.e., they define $\delta(s_h, a_h) = \mathrm{concat}[s_h, a_h]$ (Qu et al., 2024; Xiong et al., 2024). However, in iterative refinement tasks, the most recent answer and its feedback are more crucial than earlier responses, as the actor no longer needs to address errors that have already been corrected in previous iterations. Based on this heuristic, we design $\pi_a$ to refine its previous answer while observing only its most recent response and the corresponding feedback. Likewise, the critic provides feedback exclusively based on the latest answer. Formally, function $\delta$ is defined as $\delta(s_h, a_h) := [\boldsymbol{x}, a_h]$ if $h$ is even, else $\mathrm{concat}[s_h, a_h]$.

This design simplification not only reduces the model's context length requirements but also enables generalization to longer test-time horizons than train-time horizon. Intuitively, if the actor and critic are trained to effectively improve diverse initial answers, the two-agent system can iteratively refine an answer multiple times, even if it was only trained for a single refinement step. Since the actor and critic only observe the last answer, future refinement iterations are not affected by test-time horizon shift. Therefore, our practical version of DPSDP involves only one refinement step at train-time, resulting in a compact MDP with horizon $H = 3$. Our evaluation results in Section 4.3 demonstrate that DPSDP policies are effective to improve answer over turns regardless of the horizon difference, while the reduction in horizon enables easy and efficient implementation.

**Estimation of $Q$-Values** Algorithm 1 assumes access to the $Q$-value function $Q_h^{\widehat{\pi}_{h+1}}(s_h, a_h)$, which is impractical. In practice, we approximate the $Q$-value by rolling out the policy and evaluating the correctness of its answers. Particularly, for the refined answer $a_2$, its unbiased estimated $Q$-value is naturally given by its own correctness, $\widetilde{Q}_2^{\widehat{\pi}_3}(s_2, a_2) = r(s_3)$. For the feedback $a_1$, we use the reference policy $\pi_{\mathsf{ref}}$ as an approximation of $\widehat{\pi}_2$ to generate a refined answer $a_2'$ based on $a_1$, and estimate the expected return using the correctness of $a_2'$, i.e.,

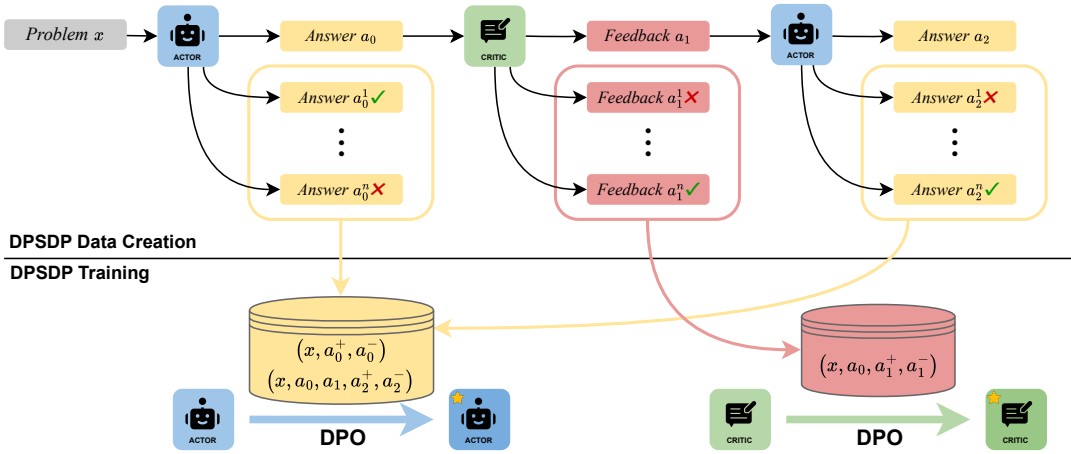

*Figure 2.* Model training. DPSDP first samples a complete trajectory $\tau = (\boldsymbol{x}, a_0, a_1, a_2)$ from the reference policy $\pi_{\mathsf{ref}}$. At each state along this trajectory, it generates $n$ responses to explore possible answers and feedback. $Q$-values of these $n$ candidate responses are then estimated as in Section 3.3 and a pairwise preference dataset is extracted for subsequent DPO training on both the actor and critic.

$\widetilde{Q}_1^{\widehat{\pi}_2}(s_1, a_1) = \mathbb{E}_{a_2' \sim \pi_{\mathsf{ref}}(\cdot|s_2)}[r(\delta(s_2, a_2'))]$. Since we focus on KL-regularized target in Eq. (1), $\pi_{\mathsf{ref}}$ is a proper approximation of $\widehat{\pi}_2$, and our results in Appendix E.3 also supports our approach. For the first-turn answer $a_0$, we estimate $Q_0^{\widehat{\pi}_1}(s_0, a_0)$ based on its correctness, i.e., $\widehat{Q}_0^{\widehat{\pi}_1}(s_0, a_0) \propto r(s_1) = \mathbb{I}[a_0 = \boldsymbol{a}^*]$. This approach is motivated by three key considerations: **(1)** in an ideal scenario where the policy from the last iteration is optimal for all actions after $h = 0$, i.e., $\widehat{\pi}_1 = \pi^*$, then $Q_0^{\widehat{\pi}_1}(s_0, a_0) = Q_0^{\pi^*}(s_0, a_0) = r(s_1) + \frac{H-1}{2}$, which differs from $r(s_1)$ by only a constant shift. **(2)** we want to encourage correct first-turn answers, as they serve as the foundation for subsequent refinements. **(3)** empirical results indicate that only a small fraction of first-turn answers change correctness in later turns relative to the total number of problems. This suggests a strong positive correlation between $r(s_1)$ and $Q_0^{\widehat{\pi}_1}(s_0, a_0)$. We analyze the impact of $Q$-value estimation on the performance guarantee in Appendix C.3.

Remarkably, all estimated $Q$-values can be obtained by rolling out $\pi_{\mathsf{ref}}$, which effectively eliminates the loop-carried dependencies across different $h$ in Algorithm 1. This allows us to consolidate all pairwise datasets into a single dataset, $\mathcal{D}_{\mathcal{H}} = \bigcup_{i=0}^{H-1} \mathcal{D}_h$, and perform a single optimization step.

**Reduction of Cross-Entropy Loss to DPO Loss**  In practice, experiments typically use $\beta \approx 0.1$, which may be significantly larger than the value suggested by Theorem 1: According to Lemma C.5 of Xie et al. (2024), $\varepsilon_{\mathsf{stat}}$ is of order $O(1/n)$, where $n$ is the number of samples used to solve the objective, indicating that $\beta$ should be small. However, it is important to note that the objective in Eq. (1) remains unchanged if both the $Q$-values

and $\beta$ are scaled by the same factor. Motivated by this, we amplify the estimated $Q$-values from their original values, which are either 0 or a positive constant. For each pair $(s_h, a_h^1, a_h^2)$ in Algorithm 1 where the estimated $Q$-values differ, we relabel the actions as $\{a_h^+, a_h^-\} = \{a_h^1, a_h^2\}$ such that $\widetilde{Q}_h^{\widehat{\pi}_{h+1}}(s_h, a_h^+) > 0$ and $\widetilde{Q}_h^{\widehat{\pi}_{h+1}}(s_h, a_h^-) = 0$. We increase $\widetilde{Q}_h^{\widehat{\pi}_{h+1}}(s_h, a_h^+)$ to a sufficiently large positive value while leaving $\widetilde{Q}_h^{\widehat{\pi}_{h+1}}(s_h, a_h^-)$ unchanged. By substituting the amplified $Q$-values into Eq. (2) and leveraging the fact that $\sigma(+\infty) = 1$, we find that Eq. (2) can be effectively estimated with a DPO loss:

$$
\begin{aligned}
\mathcal{L}_{\mathsf{DPO}}(\pi, \pi_{\mathsf{ref}}; \mathcal{D}) = -\mathbb{E}_{(s_h, a_h^+, a_h^-) \sim \mathcal{D}} \Bigg[ \\
\log \sigma \left( \beta \log \frac{\pi(a_h^+ \mid s_h)}{\pi_{\mathsf{ref}}(a_h^+ \mid s_h)} - \beta \log \frac{\pi(a_h^- \mid s_h)}{\pi_{\mathsf{ref}}(a_h^- \mid s_h)} \right) \Bigg].
\end{aligned}
\tag{3}
$$

This significantly simplifies the implementation of DPSDP without compromising performance.

Furthermore, we sample a complete trajectory $\tau = (s_0, a_0, a_1, \ldots, a_{H-1})$ from $\pi_{\mathsf{ref}}$ for each problem $\boldsymbol{x}$ and use the resulting states $s_h$ as an alternative to directly sampling from $d_h^{\pi_{\mathsf{ref}}}$. The practical implementation of DPSDP is presented in Algorithm 2 and illustrated in Figure 2.

### 3.4. Preliminary Training

We find a direct adoption of DPSDP on off-the-shelf models leads to inferior improvement, as the base models struggle to reflect on prior responses or refine their answers based on feedback. Aligned with prior works (Qu et al., 2024; Xiong et al., 2024), we observe it is beneficial to supervised fine-tuning base models on feedback and refined answers generated by a capable model as a preliminary

---

**Algorithm 2** Practical DPSDP

**Input:** reference policy $\pi_{\mathsf{ref}} = (\pi_{\mathsf{ref}\,a}, \pi_{\mathsf{ref}\,c})$, $\beta$, dataset $\mathcal{D}_{\mathsf{prob}}$, and number of samplings $n$

  Let $H = 3$.

  **for** $(\boldsymbol{x}, \boldsymbol{a}^*) \in \mathcal{D}_{\mathsf{prob}}$ **do**

    Sample a trajectory $\tau \sim \pi_{\mathsf{ref}}$ with $s_0 = \boldsymbol{x}$, $\tau = (s_0, a_0, a_1, a_2)$.

    **for** $h = 0, 1, \ldots, H - 1$ **do**

      Sample $n$ responses $a_h^i \sim \pi_{\mathsf{ref}}(\cdot \mid s_h)$, where $i = 1, \cdots, n$, and estimate their $Q$-values $\widetilde{Q}_h^{\widehat{\pi}}(s_h, a_h^i)$ as described in Section 3.3.

      Extract $m$ pairs $(a_h^+, a_h^-)$ from $\{a_h^i\}_{i=1}^{n}$ where $\widetilde{Q}_h^{\widehat{\pi}}(s_h, a_h^+) > \widetilde{Q}_h^{\widehat{\pi}}(s_h, a_h^-)$, collected as $\mathcal{D}_h$.

    **end for**

  **end for**

  Let $\widehat{\pi} = (\widehat{\pi}_a, \widehat{\pi}_c)$ be

$$\widehat{\pi}_a \leftarrow \min_{\pi} \mathcal{L}_{\mathsf{DPO}}(\pi, \pi_{\mathsf{ref}\,a}; \mathcal{D}_0 \cup \mathcal{D}_2)$$

$$\widehat{\pi}_c \leftarrow \min_{\pi} \mathcal{L}_{\mathsf{DPO}}(\pi, \pi_{\mathsf{ref}\,c}; \mathcal{D}_1)$$

  where $\mathcal{L}_{\mathsf{DPO}}$ is defined as Eq. (3).

**Ourput:** DPSDP policy $\widehat{\pi}$

---

training phase before DPSDP. This phase enables the actor to better utilize feedback for refinement and equips the critic to provide more insightful reflections. We defer training details of this phase to Appendix D.1.

# 4. Experimental Evaluation

## 4.1. Experiment Setup

**Tasks and Datasets** We focus on mathematical problem-solving tasks, evaluating our approach on **MATH 500** (Hendrycks et al., 2021b) and **GSM8K** (Cobbe et al., 2021) benchmarks. In general, MATH 500 is more challenging than GSM8K. We use problems from the OpenMathInstruct-2 (Toshniwal et al., 2025) for training, which are sourced or augmented from MATH and GSM8K—the same datasets used for benchmarking. To assess generalizability of our models to out-of-distribution problems, we evaluate on two additional benchmarks: **MMLU-Pro Math** (Wang et al., 2024b) and **Olympiad Bench** (He et al., 2024). Among them, the Olympiad Bench is the most challenging benchmark, featuring Olympiad-level scientific questions.

**Baselines** We compare our algorithm against methods adapted to enhance LLM-generated responses: **(1) STaR** (Zelikman et al., 2022) fine-tunes the models on self-generated correct answers. Specifically, for each problem in the training set, STaR samples multiple complete answer-feed-back-refinement trajectories and fine-tunes the actor and critic using only trajectories that produce correct final answers. During SFT, messages from the other LLM agent are masked: the actor's loss is computed solely from answers, while the critic's loss is based only on feedback. **(2) STaR-DPO** follows the same data collection strategy as STaR but incorporates both correct and incorrect answer trajectories. It applies DPO to train the actor and critic using a pairwise preference dataset derived from self-generated trajectories. Same as STaR, loss computation is restricted to the messages of the individual LLM agent, with messages from other agents masked. **(3) Oracle-RISE** (Qu et al., 2024) samples multiple sequent answers under the assumption of access to an oracle model that provides ground-truth feedback on the actor's previous response. At each refinement iteration, the actor is informed of the correctness of its last answer and then generates a new attempt. Our implementation of Oracle-RISE retrains base models starting from the preliminary training stage using binary feedback, followed by RL training. All baselines are trained on the same problem set as their DPSDP counterparts to ensure a fair comparison.

**Implementation** We conduct experiments using Ministral-8B-Instruct-2410, Llama-3.1-8B-Instruct, and Qwen2.5-3B as base models. We adopt a subset of OpenMathInstruct-2 as the problem set. For each state, we sample $n = 8$ additional answers or feedback using a temperature of $1.0$. We construct at most one chosen-rejected pair from these 8 candidate outputs using the estimated $Q$-values described in Section 3.3, and apply DPO loss with the collected dataset to train the actor and critic separately. Beyond a generative critic that provides verbal feedback, we also implement a value-based critic, referred to as *non-gen critic* $\pi^{\mathsf{NG}}$, which gives only binary correctness feedback. Under this setting, the critic is trained using cross-entropy loss, while the actor is fine-tuned to refine its answers based on binary feedback. More details of our implementation can be found in Appendix D.

## 4.2. Main Results

We evaluate our methods and present the main results in Table 1, and a more comprehensive set of results is provided in Appendix E. Following Qu et al. (2024), we use three key metrics:

- **pass1@turn1 (p1@t1)** measures the accuracy of the actor's initial response without any reflection or refinement.

- **maj1@turn5 (m1@t5)** computes the accuracy of the majority voting answer over five generated answers: the initial response followed by four refinements from the actor with feedback from critic. To mitigate the inherent randomness of LLM evaluations, unless oth-

| Approach | ID | | | | | | OOD | | | | | |
|---|---|---|---|---|---|---|---|---|---|---|---|---|
| | MATH 500 | | | GSM8K | | | MMLU-Pro Math | | | Olympiad Bench | | |
| | p1@t1 | m1@t5 | p1@t5 | p1@t1 | m1@t5 | p1@t5 | p1@t1 | m1@t5 | p1@t5 | p1@t1 | m1@t5 | p1@t5 |
| **Ministral-8B-It** | 55.8 | 53.4 | 58.4 | 83.4 | 81.9 | 84.7 | 52.1 | 50.6 | 55.7 | 22.8 | 22.7 | 24.8 |
| +SFT | 55.4 | 57.2 | 68.0 | 82.3 | 82.3 | 88.2 | 50.4 | 49.7 | 62.2 | 22.1 | 23.4 | 30.7 |
| +DPSDP (ours) | 58.2 | 63.2 | 70.0 | 87.8 | 89.1 | 92.7 | 53.1 | 54.2 | 64.3 | 25.8 | 27.0 | 32.9 |
| *Non-Gen Critic* | | | | | | | | | | | | |
| +SFT | 57.0 | 59.0 | 62.0 | 84.5 | 85.1 | 86.2 | 50.6 | 51.7 | 54.4 | 24.2 | 24.8 | 27.2 |
| +DPSDP | 59.2 | 62.0 | 64.2 | 88.9 | 90.4 | 91.0 | 52.8 | 53.3 | 57.8 | 25.8 | 26.6 | 30.0 |
| *Single-Agent* | 57.0 | 59.6 | 63.8 | 89.9 | 90.4 | 91.2 | 54.0 | 54.9 | 58.0 | 23.6 | 26.3 | 28.2 |
| *Non-Markovian* | 58.2 | 62.6 | 67.8 | 87.8 | 88.9 | 91.3 | 53.1 | 52.8 | 59.7 | 25.8 | 26.4 | 29.5 |
| **Llama-3.1-8B-It** | 49.2 | 43.2 | 60.2 | 83.4 | 67.0 | 88.5 | 50.1 | 41.7 | 62.0 | 18.1 | 14.8 | 26.6 |
| +SFT | 51.4 | 53.4 | 56.8 | 83.3 | 78.9 | 86.6 | 55.7 | 56.1 | 61.4 | 20.5 | 22.3 | 24.6 |
| +DPSDP (ours) | 55.8 | 58.4 | 62.0 | 87.5 | 88.4 | 91.2 | 56.6 | 58.0 | 62.1 | 22.4 | 23.0 | 25.1 |
| *Non-Gen Critic* | | | | | | | | | | | | |
| +SFT | 53.2 | 54.2 | 57.4 | 84.2 | 85.1 | 87.1 | 55.4 | 55.8 | 60.0 | 20.5 | 21.7 | 23.3 |
| +DPSDP | 56.0 | 56.2 | 60.2 | 88.6 | 89.8 | 91.2 | 56.9 | 56.5 | 62.1 | 20.2 | 20.6 | 23.1 |
| *Single-Agent* | 53.4 | 54.8 | 58.0 | 87.9 | 87.6 | 90.4 | 56.1 | 57.3 | 62.0 | 23.0 | 21.5 | 25.1 |
| *Non-Markovian* | 55.8 | 57.2 | 61.2 | 87.5 | 88.2 | 91.0 | 56.6 | 57.0 | 60.5 | 22.4 | 22.6 | 24.6 |
| **Qwen2.5-3B** | 57.6 | 48.0 | 58.6 | 78.6 | 75.2 | 79.4 | 47.4 | 41.2 | 48.4 | 24.0 | 22.0 | 24.5 |
| +SFT | 60.0 | 60.4 | 64.6 | 79.1 | 77.7 | 81.5 | 50.9 | 51.4 | 56.0 | 23.9 | 24.8 | 26.4 |
| +DPSDP (ours) | 60.4 | 62.0 | 65.2 | 79.9 | 79.9 | 84.2 | 52.6 | 53.2 | 57.1 | 24.0 | 24.0 | 26.0 |
| **STaR** | | | | | | | | | | | | |
| Ministral-8B-It | 54.6 | 55.0 | 64.6 | 84.9 | 85.7 | 89.9 | 47.2 | 47.0 | 58.6 | 21.5 | 22.7 | 29.4 |
| Llama-3.1-8B-It | 50.8 | 52.2 | 56.8 | 83.6 | 81.3 | 87.5 | 53.8 | 54.6 | 58.5 | 20.5 | 20.3 | 22.4 |
| Qwen2.5-3B | 59.0 | 59.6 | 64.8 | 80.3 | 79.5 | 83.7 | 51.9 | 51.8 | 56.8 | 23.3 | 22.6 | 24.8 |
| **STaR-DPO** | | | | | | | | | | | | |
| Ministral-8B-It | 56.6 | 58.8 | 67.8 | 87.6 | 89.5 | 92.7 | 51.6 | 52.6 | 63.2 | 25.2 | 26.4 | 31.8 |
| Llama-3.1-8B-It | 54.2 | 55.6 | 59.2 | 87.5 | 87.4 | 90.3 | 54.8 | 55.0 | 60.3 | 20.9 | 21.5 | 24.3 |
| Qwen2.5-3B | 60.4 | 60.2 | 64.8 | 79.4 | 78.9 | 82.6 | 51.2 | 52.3 | 55.9 | 23.1 | 22.8 | 28.9 |
| **Oracle-RISE** | | | | | | | | | | | | |
| Ministral-8B-It | 59.2 | 65.4 | 65.8 | 88.9 | 92.6 | 92.9 | 52.8 | 61.3 | 62.4 | 25.8 | 30.6 | 30.9 |

*Table 1.* Performance comparison of various approaches. DPSDP effectively improves reasoning performance from supervised fine-tuned models and enables agents to achieve higher accuracies by generating more answers. The highest accuracies achieved by the same base models (excluding Oracle-RISE, which has access to the ground-truth answers) under each metric are underlined.

erwise specified, questions with no more than two correct responses are considered incorrect.

- **pass1@turn5 (p1@t5)** is also based on five generated answers but considers a problem solved if at least one answer is correct. This metric is particularly valuable in settings where an oracle or reward model is available to verify correctness.

The effectiveness of our algorithm is demonstrated in Table 1. For instance, DPSDP improves the five-turn majority voting accuracy of the Ministral-based model from 57.2% to 63.2% on MATH 500 and from 82.3% to 89.1% on GSM8K compared to the models after preliminary training

phase. By rolling out five answers, DPSDP policy enables the actor to increase the correctness of its first answer by 5.0% on MATH 500 and 1.3% on GSM8K. In contrast, supervised fine-tuned models achieve only a 1.8% improvement on MATH 500 and show no gain on GSM8K.

**Comparison with Baselines** We implement the STaR and STaR-DPO baselines using the supervised fine-tuned models obtained during the DPSDP training pipeline. Interestingly, STaR, which does not incorporate incorrect final answers during training, leads to a negligible improvement with five refinement iterations. STaR-DPO achieves near-saturated performance on GSM8K but offers only a

modest accuracy gain over the first-turn answer on the more challenging MATH 500 benchmark (e.g., 2.2% for Ministral-based models). The contrast between STaR and STaR-DPO highlights the importance of incorporating negative data during training, which helps prevent response degradation in subsequent attempts. While STaR-DPO demonstrates the potential for iterative answer refinement, its lack of the restarting mechanism used in DPSDP limits its train-time exploration. On Ministral-based models, this results in a five-turn accuracy of 58.8%, which is notably lower than the 63.2% achieved by DPSDP. This gap is also evident in Llama- and Qwen-based models. We further instantiate Oracle-RISE using Ministral base models. Our results show that DPSDP models achieve maj@t5 accuracies approaching those of Oracle-RISE on challenging benchmarks such as MATH 500 and Olympiad Bench, despite Oracle-RISE having access to ground-truth feedback. Notably, our models consistently outperform Oracle-RISE on pass@t5, suggesting that the actor—guided by critic feedback—explores the solution space more actively instead of relying solely on initial responses.

**Comparison with Self-Consistency** Wang et al. (2022) highlight the effectiveness of majority voting over multiple sampled responses. A potential concern is that the improvement in maj1@t5 may stem from sampling multiple answers rather than the cooperative refinement of multi-agent interactions. To investigate this, we evaluate the actors on maj5@t1 (m5@t1), which measures the accuracy of majority voting over five independently sampled first-turn answers. As shown in Table 2, the results confirm that the performance gains primarily come from the models' ability to identify and correct errors from previous responses rather than from majority voting alone—particularly on challenging benchmarks like MATH 500.

| DPSDP | MATH 500 | | | GSM8K | | |
|---|---|---|---|---|---|---|
| | p1@t1 | m1@t5 | m5@t1 | p1@t1 | m1@t5 | m5@t1 |
| **Ministral** | 58.2 | 63.2 | 59.2 | 87.8 | 89.1 | 88.9 |
| **Llama-3.1** | 55.8 | 58.4 | 55.2 | 87.5 | 88.4 | 88.2 |

*Table 2.* Comparison between m1@t5 and m5@t1. The performance gains stem mainly from the model's ability to identify and correct errors, rather than relying solely on majority voting.

**Generalization to Out-of-Distribution (OOD) Problems** Since DPSDP is trained exclusively on problems from OpenMathInstruct-2—an augmented collection of MATH and GSM8K problems—a key question is whether it can generalize to unseen benchmarks. To assess this, we evaluate our models on MMLU-Pro Math and Olympiad Bench. The results show, even on challenging Olympiad-

level problems, Ministral-base models improve their first-turn answers over five attempts by 1.2%, demonstrating that they have internalized the ability to iteratively refine responses rather than merely memorizing in-distribution data. STaR-DPO also exhibits some generalization to OOD benchmarks, but its limited training-time exploration results in suboptimal performance compared to DPSDP.

**Generative Critic vs. Non-Generative Critic** In addition to using a generative LLM as the critic, we also experimented with a value-based reward model, where a value head is applied over the transformer's outputs to serve as the critic. We denote this non-generative critic as $\pi_c^{\mathsf{NG}}$. Given a problem $\boldsymbol{x}$ with a ground truth answer $\boldsymbol{a}^*$ and an actor-generated answer $a_h$, the non-generative critic produces a probability estimate $p_{h+1} \sim \pi_c^{\mathsf{NG}}(\cdot \mid [\boldsymbol{x}, a_h])$ between 0 and 1, approximating the probability that $a_h$ matches $\boldsymbol{a}^*$. If $p_{h+1} > 0.5$, the critic provides feedback $a_{h+1}$ affirming the previous answer; otherwise, it warns the actor that the response may contain an error. Unlike the generative critic, $\pi_c^{\mathsf{NG}}$ does not provide detailed feedback beyond correctness estimation. To evaluate the effectiveness of $\pi_c^{\mathsf{NG}}$, we train the critic alongside a corresponding actor $\pi_a^{\mathsf{NG}}$, which is specialized in processing binary feedback, forming the system $\pi^{\mathsf{NG}} = (\pi_a^{\mathsf{NG}}, \pi_c^{\mathsf{NG}})$. Our results show that DPSDP can be effectively applied in scenarios where only binary feedback is available, exemplified by the five turn accuracy improvement in Table 1. It performs particularly well on elementary problems like GSM8K, even surpassing the generative critic in this setting. We suspect this is due to the over-thinking phenomenon, where the generative critic tends to overanalyze less challenging problems, leading to performance degradation over additional turns, as observed in prior works (Chen et al., 2024a; Shridhar et al., 2024). We also provide qualitative analysis supporting this hypothesis in Appendix E.4. However, the limited feedback space of non-generative critic constrains its ability to achieve the same level of refinement as the generative version on more challenging benchmark MATH 500 and OOD benchmarks. Notably, $\pi^{\mathsf{NG}}$ consistently underperforms the generative critic in pass1@turn5, suggesting that it fails to sufficiently explore the solution space at inference time and makes less modifications to its previous answers.

### 4.3. Ablation Study and Discussion

**Single-Agent vs. Multi-Agent** We also explore whether utilizing multiple LLM agents with specialized roles contributes to the performance advantage. To investigate this, we replicate the training process on a single base model and report the evaluation results in Table 1. Our findings indicate that DPSDP effectively enhances performance across most benchmarks even with a single agent, enabling it to

both solve problems and reflect on its responses. For instance, the Ministral-based single-agent policy improves its first-turn accuracy from 57.0% to a majority voting accuracy of 59.6% on MATH 500. However, the single-agent approach underperforms compared to multi-agent policies on more challenging benchmarks such as MATH 500 and Olympiad Bench. Interestingly, we find that the single-agent framework has a slight advantage over the multi-agent approach on benchmarks containing grade-school-level problems, such as GSM8K and MMLU-Pro Math. This suggests a potential performance boost through *positive transfer* between the two related tasks of answer generation and feedback provision. For example, the actor may internalize reflection skills and preemptively avoid errors in its initial response, while the critic could enhance its feedback by learning from the answer-generation process. These observations align with prior research showing that integrating both tasks into the training objective can be beneficial (Zhang et al., 2024a; Yang et al., 2024b). We view this as a promising avenue for future research.

**Markovian vs. Non-Markovian** As discussed in Section 3.3, we define the states to incorporate only the most recent answer with the heuristic assumption that the last answer has a more profound impact on future refinements, which corresponds to the Markovian setting. The main advantage of this design choice is that our models are less affected by distribution shift, and the policy produced by DPSDP can be used to refine answers arbitrarily many times without the constraint of training-time horizon. To support our argument, we reevaluate the models with each agent having full access to previous messages, which violates the defined MDP and which we denote as the *non-Markovian* setting. Results in Table 1 show that, under non-Markovian scenarios, the policy can still improve its first-turn answer through iterative refinement. However, despite having more information available, agents suffer from test-time distribution shift and underperform compared to agents seeing only the last answer. One intuitive reason why the non-Markovian setting underperforms is that in Assumption 1, $C_{\mathcal{S}}^{\star}$ represents trajectory-level distribution shift in the non-Markovian setting. While if the Markov assumption holds, Markovian setting has state-level distribution shift for $C_{\mathcal{S}}^{\star}$ under our defined state, which is always strictly smaller than the trajectory-level shift.

**Restart vs. Non-Restart Data Collection** The data collection process in Algorithm 2 follows a restart-style approach: we sample a complete trajectory to estimate $s_h \sim d_h^{\pi_{\text{ref}}}$ and generate multiple responses from this state. We argue that this strategy enhances action space exploration by enabling more diverse responses from each sampled state $s_h$. To test this hypothesis, we modify the data collection method to a trajectory-level approach, where for each

problem $\boldsymbol{x}$, we sample $n$ full trajectories $\tau \sim \pi_{\text{ref}}(\cdot \mid \boldsymbol{x})$ and construct the DPO dataset accordingly. As shown in Table 3, our results indicate that incorporating the restart mechanism generally have a comparable or more substantial improvement performance across multiple answering attempts, supporting the idea that restart-style data collection facilitates better exploration.

| Approach | MATH 500 | | | GSM8K | | |
|---|---|---|---|---|---|---|
| | p1@t1 | m1@t5 | p1@t5 | p1@t1 | m1@t5 | p1@t5 |
| **Ministral** | | | | | | |
| w/o restart | 56.6 | 61.6 | 68.0 | 88.9 | 89.5 | 92.9 |
| w/ restart | 58.2 | 63.2 | 70.0 | 87.8 | 89.1 | 92.7 |
| **Llama-3.1** | | | | | | |
| w/o restart | 56.2 | 57.6 | 61.4 | 87.9 | 88.3 | 90.1 |
| w/ restart | 55.8 | 58.4 | 62.0 | 87.5 | 88.4 | 91.2 |

*Table 3.* Comparison of data collection with and without restart.

**Additional Analyses and Insights** In addition to our main findings, we provide extensive analyses in Appendix E to further validate the effectiveness of our approach. We examine how performance evolves as the number of refinement iterations increases, scaling up to 10 iterations, and analyze the corresponding transitions in response correctness. We also ablate the preliminary training phase to demonstrate its necessity prior to reinforcement learning. Furthermore, we empirically validate the $Q$-value estimation methods described in Section 3.3. A comprehensive qualitative analysis, including common failure patterns, is also presented. These results offer deeper insights into the mechanisms behind the observed performance gains and highlight the key design choices that contribute to our method's success.

## 5. Conclusion and Future Research Direction

We introduce DPSDP, a practical RL algorithm for multi-agent iterative solution refinement. Our algorithm offers both strong theoretical performance guarantees and substantial empirical improvements.

A promising direction for future research is developing an online or iterative algorithm that adapts to changes in state distribution during training, analogous to prior work on online/iterative DPO (Guo et al., 2024a; Dong et al., 2024). Furthermore, it is possible to explore a mixed generation objective for agents for transfer learning, as detailed in Section 4.3.

## Impact Statement

This paper presents work whose goal is to advance the field of Machine Learning. There are many potential societal consequences of our work, none which we feel must be specifically highlighted here.

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

# Appendix

## A. Related Work

A wide range of prior research focuses on improving LLM performance by enabling models to revise their answers for greater accuracy and robustness. Within this domain, a subset of works investigates *intrinsic* self-correction, where LLMs refine their outputs without relying on external feedback (Qu et al., 2024; Kumar et al., 2024). While these methods equip models with abilities to address their own errors, studies reveal mixed results, with some highlighting the difficulties LLMs face when self-correcting without external input (Kamoi et al., 2024; Huang et al., 2024). In contrast, other works leverage external feedback to enhance responses, utilizing tools such as compilers and verifiers (Welleck et al., 2022; Havrilla et al., 2024; Chen et al., 2024a; Shinn et al., 2023). More recently, multi-agent systems have gained attention as a promising approach, training two or more LLMs to collaborate or compete to generate improved solutions for reasoning tasks (Xi et al., 2024; Motwani et al., 2024). Our work aligns with this direction, proposing a novel reinforcement learning algorithm to enhance the interaction and coordination between models.

**Intrinsic Self-Correction**   Several prior works have developed techniques to enable LLMs to correct errors in their previous responses. Some approaches prompt LLMs to reflect on their answers and attempt revisions when necessary (Shinn et al., 2023; Kim et al., 2023; Madaan et al., 2023). Among these, Kim et al. (2023); Shinn et al. (2023) assume access to ground-truth answers during self-correction, which is often unrealistic. Other works explore algorithms for training models to enable models of self-correction. Kumar et al. (2024) demonstrate that standard supervised fine-tuning is insufficient for improving LLMs' ability to self-correct. Instead, they formulate self-correction as a Markov Decision Process (MDP) and apply a two-stage reinforcement learning algorithm to enhance performance, which focuses on single refinement. Qu et al. (2024) train models to iteratively refine their responses over multiple turns, which is relevant to our work. In contrast, our approach explores leveraging flexible feedback from another model to guide self-correction under this multi-turn setting.

**Self-Correction with External Feedback**   Another line of research explores self-correction using additional feedback from the environment. A common setting is code generation, where feedback is derived from unit test results or compiler messages (Chen et al., 2024b; Jain et al., 2024; Olausson et al., 2024; Ni et al., 2023). Other works incorporate external tools, such as scripts or search engines, to provide feedback for refining responses (Gou et al., 2024; Zhang et al., 2024b). Additionally, some approaches utilize feedback generated by other models (Welleck et al., 2022; Havrilla et al., 2024). However, these works treat the model generating answers and the entity providing feedback as two separate components, focusing either on leveraging feedback from a fixed set of sources (Olausson et al., 2024; Ni et al., 2023; Gou et al., 2024) or training a dedicated corrector model for refinement (Welleck et al., 2022). In contrast, our algorithm trains a multi-agent system jointly, enabling seamless interaction and collaboration between models.

**Multi-Agent Collaboration**   LLM-based multi-agent systems have shown promising results across a variety of tasks (Guo et al., 2024b; Gao et al., 2024a; Liu et al., 2024a; Jiang et al., 2024). The communication paradigms in these systems can be broadly classified into two categories (Guo et al., 2024b): **(1)** Debate or competitive agents, which interact by defending, and critiquing viewpoints or solutions (Junprung, 2023; Du and Zhang, 2024; Lan et al., 2024), and **(2)** Cooperative agents, which work together toward shared goals, exchanging information to enhance collective solutions (Hong et al., 2024b; Hang et al., 2024; Tang et al., 2024a; Wu et al., 2024b). In the context of reasoning, Du et al. (2023); Liang et al. (2023) explored improving LLMs' reasoning capabilities by employing debate-like interactions among agents. Other works (Chen et al., 2024a; Shinn et al., 2023; Xi et al., 2024; Motwani et al., 2024) refined LLM-generated responses through collaborative multi-agent frameworks, often involving specialized roles such as actor models for response generation and critic models for evaluation and feedback. However, some of these approaches lack specialized model training (Akyürek et al., 2023; Chen et al., 2024a; Shinn et al., 2023) or omit joint training of agents for collaboration (Xi et al., 2024). The most related work is Motwani et al. (2024), which trains a three-model system to generate refined answers after an initial attempt. In contrast, our work focuses on an algorithm with a theoretical performance guarantee that enables multi-turn refinement, fully encompassing the 2-turn cases.

**RL for LLMs**   Many prior works have applied reinforcement learning (RL) algorithms to LLM training. One line of research focuses on single-turn settings, where response generation is formulated as a token-level Markov Decision Process (MDP), and pairwise preference data is used to align models with human preferences (Rafailov et al., 2024; Gao et al.,

2024b; Zhong et al., 2024). Among these, Direct Preference Optimization (DPO) (Rafailov et al., 2023) demonstrates that LLMs can be aligned with human preferences by maximizing the margin between the implicit rewards of chosen and rejected responses. Building on this, many algorithms have been proposed to address specific limitations of DPO (Tang et al., 2024b; Azar et al., 2023). Some prior works explore online or iterative RL algorithms, providing theoretical performance guarantees through either high coverage of the reference policy or active exploration (Xiong et al., 2023; Guo et al., 2024a; Xu et al., 2024; Pang et al., 2024; Mitra et al., 2024; Dong et al., 2024; Xie et al., 2024). Another subset of these works focuses on a extended problem of general preference learning where binary labels are augmented to numeric probabilities representing the likelihood of one response being preferred over another (Wu et al., 2024a; Swamy et al., 2024; Munos et al., 2024; Rosset et al., 2024). Other aspects include on-policy training (Tajwar et al., 2024; Liu et al., 2024b), direct utility function maximization (Ethayarajh et al., 2024), and and simplified algorithms that eliminate the need for a reference policy (Hong et al., 2024a; Meng et al., 2024).

Another line of research focuses on improving LLM performance in multi-turn settings, where the prompt and conversation history are treated as the state, and each response is an action. Some works aim to generally align models with human preferences in multi-turn conversations (Shani et al., 2024; Zhou et al., 2024), while others specifically enhance step-by-step reasoning (Snell et al., 2025; OpenAI, 2024; Wang et al., 2023) or integrate tool usage (Xiong et al., 2024). Qu et al. (2024); Kumar et al. (2024) are most related to our work. They apply RL algorithms to train models for self-correction through multiple sequential attempts. In comparison, we formulate the conversation as each state contains only most recent answer, mitigating test-time distribution shift effect regarding number of refinements.

# B. Additional Notation and Pseudocode

We provide the pseudocode for original PSDP algorithm described in Section 2.

---

**Algorithm 3** PSDP

---

**Input:** horizon $H$, policy set $\Pi$, and baseline distributions $\mu_h$ for $h = 0, \cdots, H - 1$

1: **for** $h = H - 1, H - 2, \ldots, 0$ **do**
2:   $\pi_h \leftarrow \arg\max_{\pi \in \Pi} \mathbb{E}_{s \sim \mu_h} \left[ V_h^{(\pi, \pi^{h+1}, \cdots, \pi^{H-1})}(s) \right]$
  where $(\pi, \pi^{h+1}, \cdots, \pi^{H-1})$ denotes a non-stationary policy composed of stationary policies $\pi, \pi^{h+1}, \cdots, \pi^{H-1}$.
3: **end for**

---

# C. Theoretical Proofs

## C.1. Proof of Theorem 1

We first introduce an episodic version of performance difference lemma (Kakade and Langford, 2002).

**Lemma 2** (Performance difference lemma). *For any policy $\pi'$ and $\pi$, we have*

$$\mathcal{J}(\pi') - \mathcal{J}(\pi) = \sum_{h=0}^{H-1} \mathbb{E}_{s_h \sim d_h^{\pi'}} \left[ \mathbb{E}_{a_h \sim \pi'(\cdot|s_h)} \left[ Q_h^\pi(s_h, a_h) \right] - V_h^\pi(s_h) \right]$$

Let $f_h$ be any function, then from Lemma 2 we have

$$\mathcal{J}(\pi^\star) - \mathcal{J}(\widehat{\pi}) = \sum_{h=0}^{H-1} \mathbb{E}_{s_h \sim d_h^{\pi^\star}} \left[ \mathbb{E}_{a_h \sim \pi^\star(\cdot|s_h)} \left[ Q_h^{\widehat{\pi}}(s_h, a_h) \right] - V_h^{\widehat{\pi}}(s_h) \right] \tag{4}$$

$$= \sum_{h=0}^{H-1} \mathbb{E}_{s_h \sim d_h^{\pi^\star}, a_h \sim \pi^\star(\cdot|s_h)} \left[ Q_h^{\widehat{\pi}}(s_h, a_h) - f_h(s_h, a_h) \right]$$

$$+ \sum_{h=0}^{H-1} \mathbb{E}_{s_h \sim d_h^{\pi^\star}} \left[ \mathbb{E}_{a_h \sim \pi^\star(\cdot|s_h)} \left[ f_h(s_h, a_h) \right] - \mathbb{E}_{a_h \sim \widehat{\pi}(\cdot|s_h)} \left[ f_h(s_h, a_h) \right] \right]$$

$$+ \sum_{h=0}^{H-1} \mathbb{E}_{s_h \sim d_h^{\pi^\star}, a_h \sim \widehat{\pi}(\cdot|s_h)} \left[ f_h(s_h, a_h) - Q_h^{\widehat{\pi}}(s_h, a_h) \right].$$

Let $f_h(s_h, a_h) := \beta \log \frac{\widehat{\pi}(a_h|s_h)}{\pi_{\mathsf{ref}}(a_h|s_h)} - c(s_h)$, where $c(s_h) = \mathbb{E}_{a_h \sim \pi_{\mathsf{ref}}(\cdot|s_h)} \left[ \beta \log \frac{\widehat{\pi}(a_h|s_h)}{\pi_{\mathsf{ref}}(a_h|s_h)} - Q_h^{\widehat{\pi}}(s_h, a_h) \right]$, then

$$\mathcal{J}(\pi^\star) - \mathcal{J}(\widehat{\pi}) = \sum_{h=0}^{H-1} \mathbb{E}_{s_h \sim d_h^{\pi^\star}, a_h \sim \pi^\star(\cdot|s_h)} \left[ Q_h^{\widehat{\pi}}(s_h, a_h) - \beta \log \frac{\widehat{\pi}(a_h \mid s_h)}{\pi_{\mathsf{ref}}(a_h \mid s_h)} + c(s_h) \right] \tag{I}$$

$$+ \beta \sum_{h=0}^{H-1} \mathbb{E}_{s_h \sim d_h^{\pi^\star}} \left[ \mathbb{E}_{a_h \sim \pi^\star(\cdot|s_h)} \left[ \log \frac{\widehat{\pi}(a_h \mid s_h)}{\pi_{\mathsf{ref}}(a_h \mid s_h)} \right] - \mathbb{E}_{a_h \sim \widehat{\pi}(\cdot|s_h)} \left[ \log \frac{\widehat{\pi}(a_h \mid s_h)}{\pi_{\mathsf{ref}}(a_h \mid s_h)} \right] \right] \tag{II}$$

$$+ \sum_{h=0}^{H-1} \mathbb{E}_{s_h \sim d_h^{\pi^\star}, a_h \sim \widehat{\pi}(\cdot|s_h)} \left[ \beta \log \frac{\widehat{\pi}(a_h \mid s_h)}{\pi_{\mathsf{ref}}(a_h \mid s_h)} - Q_h^{\widehat{\pi}}(s_h, a_h) - c(s_h) \right]. \tag{III}$$

For term I, from Cauchy-Schwartz inequality, we have

$$\sum_{h=0}^{H-1} \mathbb{E}_{s_h \sim d_h^{\pi^\star}, a_h \sim \pi^\star(\cdot|s_h)} \left[ Q_h^{\widehat{\pi}}(s_h, a_h) - \beta \log \frac{\widehat{\pi}(a_h \mid s_h)}{\pi_{\mathsf{ref}}(a_h \mid s_h)} + c(s_h) \right]$$

$$\leq \sum_{h=0}^{H-1} \mathbb{E}_{s_h \sim d_h^{\pi^\star}, a_h \sim \pi^\star(\cdot|s_h)} \left[ \left| Q_h^{\widehat{\pi}}(s_h, a_h) - \beta \log \frac{\widehat{\pi}(a_h \mid s_h)}{\pi_{\mathsf{ref}}(a_h \mid s_h)} + c(s_h) \right| \right]$$

$$\leq \sqrt{H \sum_{h=0}^{H-1} \mathbb{E}_{s_h \sim d_h^{\pi^\star}, a_h \sim \pi^\star(\cdot|s_h)} \left[ \left( Q_h^{\widehat{\pi}}(s_h, a_h) - \beta \log \frac{\widehat{\pi}(a_h \mid s_h)}{\pi_{\mathsf{ref}}(a_h \mid s_h)} + c(s_h) \right)^2 \right]}$$

From Assumption 1, we have

$$\mathbb{E}_{s_h \sim d_h^{\pi^\star}, a_h \sim \pi^\star(\cdot|s_h)} \left[ \left( Q_h^{\widehat{\pi}}(s_h, a_h) - \beta \log \frac{\widehat{\pi}(a_h \mid s_h)}{\pi_{\mathsf{ref}}(a_h \mid s_h)} + c(s_h) \right)^2 \right]$$

$$\leq C_{\mathcal{S}}^\star C_{\mathcal{A}} \mathbb{E}_{s_h \sim d_h^{\pi_{\mathsf{ref}}}, a_h \sim \pi_{\mathsf{ref}}(\cdot|s_h)} \left[ \left( Q_h^{\widehat{\pi}}(s_h, a_h) - \beta \log \frac{\widehat{\pi}(a_h \mid s_h)}{\pi_{\mathsf{ref}}(a_h \mid s_h)} + c(s_h) \right)^2 \right]$$

We now introduce the following lemma to further bound term I.

**Lemma 3.** *Under Assumption 2, we have for any $h \in \{0, 1, \cdots, H-1\}$:*

$$\mathbb{E}_{s_h \sim d_h^{\pi_{\mathsf{ref}}}, a_h \sim \pi_{\mathsf{ref}}(\cdot|s_h)} \left[ \left( \beta \log \frac{\widehat{\pi}(a_h \mid s_h)}{\pi_{\mathsf{ref}}(a_h \mid s_h)} - Q_h^{\widehat{\pi}}(s_h, a_h) - c(s_h) \right)^2 \right] \leq \frac{\varepsilon_{\mathsf{stat}}}{2},$$

*where $c(s_h) = \mathbb{E}_{a_h \sim \pi_{\mathsf{ref}}(\cdot|s_h)} \left[ \beta \log \frac{\widehat{\pi}(a_h|s_h)}{\pi_{\mathsf{ref}}(a_h|s_h)} - Q_h^{\widehat{\pi}}(s_h, a_h) \right].$*

Then from Lemma 3 we know

$$\sqrt{H \sum_{h=0}^{H-1} \mathbb{E}_{s_h \sim d_h^{\pi^\star}, a_h \sim \pi^\star(\cdot|s_h)} \left[ \left( Q_h^{\widehat{\pi}}(s_h, a_h) - \beta \log \frac{\widehat{\pi}(a_h \mid s_h)}{\pi_{\mathsf{ref}}(a_h \mid s_h)} + c(s_h) \right)^2 \right]}$$

$$\leq \sqrt{C_{\mathcal{S}}^\star C_{\mathcal{A}} H \sum_{h=0}^{H-1} \mathbb{E}_{s_h \sim d_h^{\pi_{\mathsf{ref}}}, a_h \sim \pi_{\mathsf{ref}}(\cdot|s_h)} \left[ \left( Q_h^{\widehat{\pi}}(s_h, a_h) - \beta \log \frac{\widehat{\pi}(a_h \mid s_h)}{\pi_{\mathsf{ref}}(a_h \mid s_h)} + c(s_h) \right)^2 \right]}$$

$$\leq H \sqrt{\frac{C_{\mathcal{S}}^\star C_{\mathcal{A}} \varepsilon_{\mathsf{stat}}}{2}}$$

Similarly, term III can be bounded as

$$
\sum_{h=0}^{H-1} \mathbb{E}_{s_h \sim d_h^{\pi^\star}, a_h \sim \widehat{\pi}(\cdot | s_h)} \left[ \beta \log \frac{\widehat{\pi}(a_h \mid s_h)}{\pi_{\text{ref}}(a_h \mid s_h)} - Q_h^{\widehat{\pi}}(s_h, a_h) - c(s_h) \right]
$$

$$
\leq \sqrt{H \sum_{h=0}^{H-1} \mathbb{E}_{s_h \sim d_h^{\pi^\star}, a_h \sim \widehat{\pi}(\cdot | s_h)} \left[ \left( \beta \log \frac{\widehat{\pi}(a_h \mid s_h)}{\pi_{\text{ref}}(a_h \mid s_h)} - Q_h^{\widehat{\pi}}(s_h, a_h) - c(s_h) \right)^2 \right]}
$$

$$
\leq \sqrt{C_{\mathcal{S}}^\star C_{\mathcal{A}} H \sum_{h=0}^{H-1} \mathbb{E}_{s_h \sim d_h^{\pi_{\text{ref}}}, a_h \sim \pi_{\text{ref}}(\cdot | s_h)} \left[ \left( \beta \log \frac{\widehat{\pi}(a_h \mid s_h)}{\pi_{\text{ref}}(a_h \mid s_h)} - Q_h^{\widehat{\pi}}(s_h, a_h) - c(s_h) \right)^2 \right]}
$$

$$
\leq H \sqrt{\frac{C_{\mathcal{S}}^\star C_{\mathcal{A}} \varepsilon_{\text{stat}}}{2}}
$$

Now, let's consider term II. Given a fixed $h$ and fixed $s_h$,

$$
\mathbb{E}_{y \sim \pi^\star(\cdot | s_h)} \left[ \log \frac{\widehat{\pi}(a_h \mid s_h)}{\pi_{\text{ref}}(a_h \mid s_h)} \right] - \mathbb{E}_{a_h \sim \widehat{\pi}(\cdot | s_h)} \left[ \log \frac{\widehat{\pi}(a_h \mid s_h)}{\pi_{\text{ref}}(a_h \mid s_h)} \right]
$$

$$
= \mathbb{E}_{a_h \sim \pi^\star(\cdot | s_h)} \left[ \log \frac{\widehat{\pi}(a_h \mid s_h)}{\pi^\star(a_h \mid s_h)} \right] + \mathbb{E}_{a_h \sim \pi^\star(\cdot | s_h)} \left[ \log \frac{\pi^\star(a_h \mid s_h)}{\pi_{\text{ref}}(a_h \mid s_h)} \right] - \mathbb{E}_{a_h \sim \widehat{\pi}(\cdot | s_h)} \left[ \log \frac{\widehat{\pi}(a_h \mid s_h)}{\pi_{\text{ref}}(a_h \mid s_h)} \right]
$$

$$
= \mathbb{E}_{a_h \sim \pi^\star(\cdot | s_h)} \left[ \log \frac{\pi^\star(a_h \mid s_h)}{\pi_{\text{ref}}(a_h \mid s_h)} \right] - D_{\text{KL}}[\pi^\star(\cdot \mid s_h) \| \widehat{\pi}(\cdot \mid s_h)] - D_{\text{KL}}[\widehat{\pi}(\cdot \mid s_h) \| \pi_{\text{ref}}(\cdot \mid s_h)]
$$

From Assumption 1 we have

$$
\mathbb{E}_{a_h \sim \pi^\star(\cdot | s_h)} \left[ \log \frac{\pi^\star(a_h \mid s_h)}{\pi_{\text{ref}}(a_h \mid s_h)} \right] - D_{\text{KL}}[\pi^\star(\cdot \mid s_h) \| \widehat{\pi}(\cdot \mid s_h)] - D_{\text{KL}}[\widehat{\pi}(\cdot \mid s_h) \| \pi_{\text{ref}}(\cdot \mid s_h)]
$$

$$
\leq \mathbb{E}_{a_h \sim \pi^\star(\cdot | s_h)} [\log C_{\mathcal{A}}] - 0 - 0
$$

$$
= \log C_{\mathcal{A}}
$$

Therefore, with $\beta \leq \frac{\sqrt{C_{\mathcal{S}}^\star C_{\mathcal{A}} \varepsilon_{\text{stat}}}}{\log C_{\mathcal{A}}}$, term II can be bounded as

$$
\beta \sum_{h=0}^{H-1} \mathbb{E}_{s_h \sim d_h^{\pi^\star}} \left[ \mathbb{E}_{a_h \sim \pi^\star(\cdot | s_h)} \left[ \log \frac{\widehat{\pi}(a_h \mid s_h)}{\pi_{\text{ref}}(a_h \mid s_h)} \right] - \mathbb{E}_{a_h \sim \widehat{\pi}(\cdot | s_h)} \left[ \log \frac{\widehat{\pi}(a_h \mid s_h)}{\pi_{\text{ref}}(a_h \mid s_h)} \right] \right] \leq \beta H \log C_{\mathcal{A}} = H \sqrt{C_{\mathcal{S}}^\star C_{\mathcal{A}} \varepsilon_{\text{stat}}}
$$

In conclusion,

$$
\mathcal{J}(\pi^\star) - \mathcal{J}(\widehat{\pi}) = O\left( H \sqrt{C_{\mathcal{S}}^\star C_{\mathcal{A}} \varepsilon_{\text{stat}}} \right)
$$

## C.2. Proofs for Supporting Lemmas

***Proof of Lemma 2*** Let us prove by induction. When $H = 1$, the trajectory consists of a single step, and the statement is trivial.

Now assume that for horizon $k$ the statement holds for any policy $\pi'$ and $\pi$:

$$
\mathcal{J}_k(\pi') - \mathcal{J}_k(\pi) = \sum_{h=1}^{k} \mathbb{E}_{s_h \sim d_h^{\pi'}, a_h \sim \pi'(\cdot | s_h)} [Q_h^\pi(s_h, a_h) - V_h^\pi(s_h)].
$$

For $H = k + 1$, we expand $\mathcal{J}_{k+1}(\pi') - \mathcal{J}_{k+1}(\pi)$:

$$
\mathcal{J}_{k+1}(\pi') - \mathcal{J}_{k+1}(\pi) = \mathbb{E}_{\pi'} \left[ \sum_{h=1}^{k+1} r(s_h, a_h) \right] - \mathbb{E}_{\pi} \left[ \sum_{h=1}^{k+1} r(s_h, a_h) \right]
$$

$$= \mathbb{E}_{\pi'} \left[ \sum_{h=2}^{k+1} r(s_h, a_h) \right] - \mathbb{E}_{\pi'}[V_2^\pi(s_2)] \tag{I}$$

$$+ \mathbb{E}_{\pi'}[r(s_1, a_1)] + \mathbb{E}_{\pi'}[V_2^\pi(s_2)] - \mathbb{E}_\pi \left[ \sum_{h=1}^{k+1} r(s_h, a_h) \right] \tag{II}$$

Term I corresponds to the difference between expected returns of $\pi'$ and $\pi$ on an MDP with horizon $k$. Apply the inductive hypothesis to term I, we have

$$\mathbb{E}_{\pi'} \left[ \sum_{h=2}^{k+1} r(s_h, a_h) \right] - \mathbb{E}'_\pi[V_2^\pi(s_2)] = \sum_{h=2}^{H} \mathbb{E}_{s_h \sim d_h^{\pi'}, a_h \sim \pi'(\cdot|s_h)} [Q_h^\pi(s_h, a_h) - V_h^\pi(s_h)]$$

For term II, we have

$$\mathbb{E}_{\pi'}[r(s_1, a_1)] + \mathbb{E}_{\pi'}[V_2^\pi(s_2)] - \mathbb{E}_\pi \left[ \sum_{h=1}^{k+1} r(s_h, a_h) \right]$$

$$= \mathbb{E}_{\pi'}[Q_1^\pi(s_1, a_1)] - \mathbb{E}_\pi[V_1^\pi(s_1)]$$

$$= \mathbb{E}_{s_1 \sim d_1^{\pi'}} \left[ \mathbb{E}_{a_1 \sim \pi'(\cdot|s_1)} [Q_1^\pi(s_1, a_1)] - V_1^\pi(s_1) \right]$$

Summing up term I and II concludes our proof.

***Proof of Lemma 3***

$$\mathbb{E}_{s_h \sim d_h^{\pi_{\mathsf{ref}}}, a_h \sim \pi_{\mathsf{ref}}(\cdot|s_h), a'_h \sim \pi_{\mathsf{ref}}(\cdot|s_h)} \left[ \left( \beta \log \frac{\widehat{\pi}(a_h \mid s_h)}{\pi_{\mathsf{ref}}(a_h \mid s_h)} - \beta \log \frac{\widehat{\pi}(a'_h \mid s_h)}{\pi_{\mathsf{ref}}(a'_h \mid s_h)} - Q_h^{\widehat{\pi}}(s_h, a_h) + Q_h^{\widehat{\pi}}(s_h, a'_h) \right)^2 \right]$$

$$= \mathbb{E}_{s_h \sim d_h^{\pi_{\mathsf{ref}}}, a_h \sim \pi_{\mathsf{ref}}(\cdot|s_h), a'_h \sim \pi_{\mathsf{ref}}(\cdot|s_h)} \Bigg[$$

$$\left( \left( \beta \log \frac{\widehat{\pi}(a_h \mid s_h)}{\pi_{\mathsf{ref}}(a_h \mid s_h)} - Q_h^{\widehat{\pi}}(s_h, a_h) - c(s_h) \right) - \left( \beta \log \frac{\widehat{\pi}(a'_h \mid s_h)}{\pi_{\mathsf{ref}}(a'_h \mid s_h)} - Q_h^{\widehat{\pi}}(s_h, a'_h) - c(s_h) \right) \right)^2 \Bigg]$$

$$= 2\mathbb{E}_{s_h \sim d_h^{\pi_{\mathsf{ref}}}, a_h \sim \pi_{\mathsf{ref}}(\cdot|s_h)} \left[ \left( \beta \log \frac{\widehat{\pi}(a_h \mid s_h)}{\pi_{\mathsf{ref}}(a_h \mid s_h)} - Q_h^{\widehat{\pi}}(s_h, a_h) - c(s_h) \right)^2 \right]$$

$$- 2\mathbb{E}_{s_h \sim d_h^{\pi_{\mathsf{ref}}}} \left[ \left( \mathbb{E}_{a_h \sim \pi_{\mathsf{ref}}(\cdot|s_h)} \left[ \beta \log \frac{\widehat{\pi}(a_h \mid s_h)}{\pi_{\mathsf{ref}}(a_h \mid s_h)} - Q_h^{\widehat{\pi}}(s_h, a_h) - c(s_h) \right] \right)^2 \right]$$

$$= 2\mathbb{E}_{s_h \sim d_h^{\pi_{\mathsf{ref}}}, a_h \sim \pi_{\mathsf{ref}}(\cdot|s_h)} \left[ \left( \beta \log \frac{\widehat{\pi}(a_h \mid s_h)}{\pi_{\mathsf{ref}}(a_h \mid s_h)} - Q_h^{\widehat{\pi}}(s_h, a_h) - c(s_h) \right)^2 \right]$$

The last step follows from the definition of $c(s_h)$. Under Assumption 2, we have

$$\mathbb{E}_{s_h \sim d_h^{\pi_{\mathsf{ref}}}, a_h \sim \pi_{\mathsf{ref}}(\cdot|s_h)} \left[ \left( \beta \log \frac{\widehat{\pi}(a_h \mid s_h)}{\pi_{\mathsf{ref}}(a_h \mid s_h)} - Q_h^{\widehat{\pi}}(s_h, a_h) - c(s_h) \right)^2 \right] \le \frac{\varepsilon_{\mathsf{stat}}}{2}$$

### C.3. Analysis of the Approximation of $Q$-Values

In this section, we analyze the $Q$-value approximation in practical DPSDP described in Section 3.3. As only one feedback and refinement step is used during training, we assume $H = 3$. Let $\widehat{\pi}$ be the resulting policy, and let $\widetilde{Q}_h^{\widehat{\pi}}$ denote the estimated $Q$-values, replacing $Q_h^{\widehat{\pi}}$ in Assumption 2.

We define advantage function as $A_h^\pi(s_h, a_h) = Q_h^\pi(s_h, a_h) - V_h^\pi(s_h)$, and $\widetilde{A}_h^\pi(s_h, a_h) = \widetilde{Q}_h^\pi(s_h, a_h) - \mathbb{E}_{a_h \sim \pi(\cdot|s_h)}[\widetilde{Q}_h^\pi(s_h, a_h)]$. Then Eq. (4) can be rewritten as

$$\mathcal{J}(\pi^\star) - \mathcal{J}(\widehat{\pi}) = \sum_{h=0}^{H-1} \mathbb{E}_{s_h \sim d_h^{\pi^\star}, a_h \sim \pi^\star(\cdot|s_h)} \left[ A_h^{\widehat{\pi}}(s_h, a_h) \right]. \tag{5}$$

We analyze each term in Eq. (5) as follows:

1. At $h = 2$, the estimated $\widetilde{Q}_2^{\hat{\pi}_3}(s_2, a_2) = r(s_3)$ is exact. Therefore, $A_h^{\hat{\pi}}(s_h, a_h) = \widetilde{A}_h^{\hat{\pi}}(s_h, a_h)$.

2. At $h = 1$, the estimated $Q$-value is: $\widetilde{Q}_1^{\hat{\pi}_2}(s_1, a_1) = \mathbb{E}_{a_2 \sim \pi_{\text{ref}}(\cdot|s_2)}[r(s_3)] = Q_1^{\pi_{\text{ref}}}(s_1, a_1)$. We define the approximation error:

$$\Delta = \mathbb{E}_{s_h \sim d_h^{\pi^\star}, a_h \sim \pi^\star(\cdot|s_h)}[A_h^{\hat{\pi}}(s_h, a_h) - \widetilde{A}_h^{\hat{\pi}}(s_h, a_h)]$$

3. At $h = 0$, we have $\widetilde{Q}_0^{\hat{\pi}_1}(s_0, a_0) = r(s_1) + \frac{H-1}{2} = Q_0^{\pi^\star}(s_0, a_0)$. Therefore,

$$\mathbb{E}_{a_h \sim \pi^\star(\cdot|s_h)}[A_h^{\hat{\pi}}(s_h, a_h)] \approx \mathbb{E}_{a_h \sim \pi^\star(\cdot|s_h)}[A_h^{\pi^\star}(s_h, a_h)] = 0,$$

where the last equality follows from the definition of $A_h^\pi$.

Following steps in Appendix C.1, we obtain the approximate upper bound by adding $|\Delta|$ to the theoretical bound.

To assess the impact of $|\Delta|$, we performed an ablation using the step-by-step DPSDP variant in Appendix E.3, which uses $Q_1^{\pi_2}$ in the DPO-style loss. The results showed no significant performance gain, indicating that $|\Delta|$ has minimal effect. For simplicity and efficiency, we use the original version in the main paper.

## D. Implementation Details

### D.1. Preliminary Training

As described in Section 3.4, we begin by supervised fine-tuning the model to initialize the critic for providing effective feedback and the actor for refining previous answers. To acquire high-quality SFT dataset, we generate first-turn responses by rolling out the reference policies, exposing errors that a learner actor might make. An oracle model then generates feedback and corresponding refined answers, demonstrating expected behavior for critic and actor. After this initial stage, we proceed with DPSDP from the boosted model.

In experiments, we use a subset of mathematical problems from OpenMathInstruct-2 (Toshniwal et al., 2025) and sample diverse first-turn answers with the base models Ministral-8B-Instruct-2410 and Llama-3.1-8B-Instruct. To generate high-quality feedback and refined answers, we employ capable models Mistral-Large-Instruct-2411 and Llama-3.3-70B-Instruct, to demonstrate how to evaluate and improve initial responses. After filtering (e.g., deduplicating problems and removing degraded answers), we obtain a final dataset of 381K samples for supervised fine-tuning. We then fine-tune the base models separately: the critic is trained on feedback generation, while the actor is trained on refining answers. Both DPSDP and baseline methods are subsequently trained on these fine-tuned models.

### D.2. Benchmarks

(1) **MATH 500** (Hendrycks et al., 2021b) consists of problems from mathematics competitions, including the AMC 10, AMC 12, AIME, etc. The dataset covers diverse mathematical topics, including algebra, geometry, statistics, number theory, linear algebra, and calculus. Following Lightman et al. (2023), we augment the MATH training set with 4500 problems from the test set and report results on the remaining 500 problems, denoted as **MATH 500**. (2) **GSM8K** (Cobbe et al., 2021) includes 1319 grade-school-level math word problems, which are generally simpler than those in MATH. (3) **MMLU-Pro Math** (Wang et al., 2024b) is a mathematical subset of MMLU-Pro, a challenging multi-task benchmark. It consists of 1351 multiple choice questions augmented from TheoremQA (Chen et al., 2023), SciBench (Wang et al., 2024a), and original MMLU (Hendrycks et al., 2021a). (4) **Olympiad Bench** (He et al., 2024) is an Olympiad-level scientific benchmark, containing 674 open-ended text-only mathematical problems in English, covering algebra, combinatorics, geometry, and number theory.

### D.3. Training Hyperparameters

For supervised fine-tuning (SFT), we experimented with learning rates of $1 \times 10^{-6}$, $5 \times 10^{-6}$, and $1 \times 10^{-5}$, selecting $1 \times 10^{-6}$ for Llama-based models and $5 \times 10^{-6}$ for Ministral- and Qwen-based models. Base models were trained on the SFT dataset for 1 epoch, using gradient accumulation steps of 64 and a per-device train batch size of 1 on 4×H100 80GB GPUs.

For direct preference optimization (DPO), we tested learning rates of $2 \times 10^{-7}$ and $4 \times 10^{-7}$, choosing $2 \times 10^{-7}$ for Ministral-based actor and critic, Llama-based actor, and Qwen-based actor, and $4 \times 10^{-7}$ for Llama- and Qwen-based critics. For the KL coefficient $\beta$, we evaluated values of 0.1, 0.5, and 1.0, selecting 0.1 for all actor model training, 1.0 for Ministral-8B-Instruct-based critic (trained for 1 epoch), and 0.1 for Llama-3.1-8B-Instruct-based critic (trained for 2 epochs) and Qwen2.5-3B-based critic (trained for 3 epochs). Same as in SFT, we used gradient accumulation steps of 64 and a per-device train batch size of 1 on 4×H100 80GB GPUs.

The non-generative critic was trained with a learning rate of $1 \times 10^{-6}$ for 1 epoch.

### D.4. Inference Hyperparameters

All evaluations were performed with a temperature ($T$) of 0, except for self-consistency (maj5@turn1), where we set $T = 0.5$. To generate dataset used in the preliminary training phase (SFT), reference models (Ministral-8B-Instruct-2410 and Llama-3.1-8B-Instruct) generated first-turn answers with $T = 0.8$. Oracle models (Mistral-Large-Instruct-2411 and Llama-3.3-70B-Instruct) then produced feedback and refined answers with $T = 0$. In the data collection stage of Algorithm 2, policies were sampled with $T = 1.0$ to ensure diverse responses.

### D.5. Codebase

We adapted code from TRL (von Werra et al., 2020) for both SFT and DPO training. The non-generative critic was trained using the Hugging Face Transformers framework (Wolf et al., 2020) with a customized loss function. For inference, we utilized the vLLM offline engine (Kwon et al., 2023) and adapted scripts from Xiong et al. (2024). Evaluation code was adapted from Yang et al. (2024a); Grattafiori et al. (2024) to compare LLM-generated answers with ground-truth solution.

### D.6. Prompts

> **Prompts used for first-turn answer feedback and refined answers**
>
> **For first-turn answer** You are an AI language model designed to assist with math problem-solving. In this task, I will provide you with math problems. Your goal is to solve the problem step-by-step, showing your reasoning at each step. After you have finished solving the problem, present your final answer as \boxed{Your Answer}. {problem}
> **For feedback** Take a moment to review your previous response for accuracy and completeness. Briefly identify any errors or gaps that might lead to incorrect answers, but don't worry about fixing them—just focus on pointing them out.
> **For refined answer** Using your reflection as a guide, carefully make any necessary corrections to your previous response. Ensure your final answer addresses any identified issues and present it as \boxed{Your Answer}. Pay close attention to the errors you pinpointed during the reflection.

> **Verbal feedback from non-generative critic**
>
> **Correct** The solution appears to be correct. Please review it to confirm its accuracy, and present the verified final answer in the format \boxed{Your Answer}.
> **Incorrect** There is an error in the solution. Please review it, correct the mistake, and present the revised answer in the format \boxed{Your Answer}.

## E. Additional Results

### E.1. Answer Improvement Over Turns

The training process of DPSDP involves single-step refinement rather than multi-turn enhancement. Despite this discrepancy between training- and test-time conditions, we find that DPSDP effectively enables models to improve accuracy over multiple turns. In Figure 3, we plot the per-turn accuracy, majority voting accuracy, and pass@t-$k$ accuracies on MATH 500 and GSM8K. For this section only, we do not consider questions with no more than two correct responses as incorrect, which is adopted in Section 4.2. The results show a general increase in accuracy as the number of refinements grows, with

models steadily improving majority voting performance over successive turns. Additionally, the increasing pass1@turn-$k$ scores indicate that models are progressively solving problems they previously could not, demonstrating the effectiveness of iterative refinement.

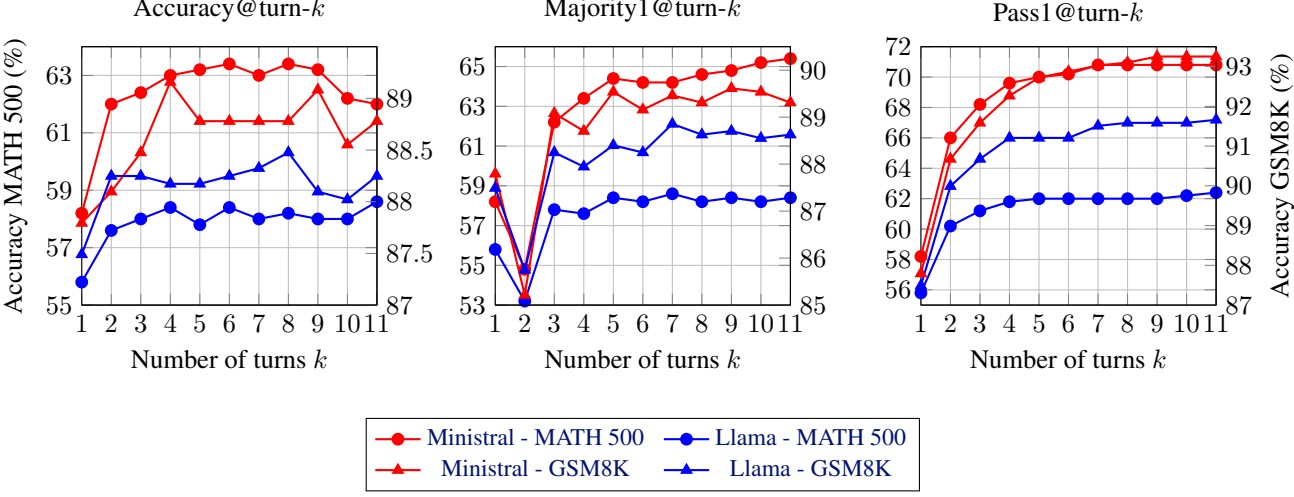

*Figure 3.* Various metrics under different turns. Accuracies improve as the number of refinements increases. The rising pass1@turn-$k$ scores indicate that iterative refinement enables models to solve previously unsolved problems. Note that the decrease in maj1@t2 accuracy arises from the requirement that both responses (2 out of 2) must be correct to count toward maj1@t2.

We further analyze how the proportion of responses that change in correctness evolves throughout the multi-turn refinement process. Table 4 reports results using Ministral-based models on the MATH 500 dataset. We focus on the fraction of responses transitioning from incorrect to correct, denoted $\Delta^{i \to c}$, and from correct to incorrect, denoted $\Delta^{c \to i}$, at each refinement step. Notably, $\Delta^{i \to c}$ consistently exceeds $\Delta^{c \to i}$, indicating that the refinement process is generally beneficial. Interestingly, both $\Delta^{i \to c}$ and $\Delta^{c \to i}$ decline over successive iterations, suggesting an initial phase of active correction followed by stabilization in later stages. The consistently low level of $\Delta^{c \to i}$ suggest that over-refinement is not a significant concern in practice. However, to further eliminate the possibilities of over-refinement, we propose monitoring performance on a validation set at each refinement step and stopping early if accuracy begins to decline as a potential solution.

| Turn | $t2$ | $t3$ | $t4$ | $t5$ | $t6$ | $t7$ | $t8$ | $t9$ | $t10$ | $t11$ |
|---|---|---|---|---|---|---|---|---|---|---|
| $\Delta^{i \to c}$ (%) | 7.8 | 4.0 | 3.4 | 3.0 | 1.8 | 2.0 | 1.4 | 1.0 | 1.0 | 1.2 |
| $\Delta^{c \to i}$ (%) | 4.0 | 3.6 | 2.8 | 2.8 | 1.6 | 2.4 | 1.0 | 1.2 | 2.0 | 1.4 |

*Table 4.* Fraction of responses that transition from incorrect to correct ($\Delta^{i \to c}$) and from correct to incorrect ($\Delta^{c \to i}$) at each turn.

### E.2. Necessity of Preliminary Training Phase

As detailed in Section 3.4, our algorithm begins with models that have been fine-tuned to either utilize or provide feedback. To assess the necessity of this preliminary training, we conduct an ablation study by removing this phase and applying DPSDP directly to base models. The results, shown in Table 5, indicate that this fine-tuning step is essential for enabling the actor and critic models to follow instructions effectively. Without it, applying DPSDP yields negligible performance gains with additional response attempts. For instance, the models achieve only 53.8% accuracy on MATH 500—an improvement of just 1.2% over a single-response baseline. In contrast, incorporating the preliminary training leads to a 5.0% absolute gain in accuracy. These findings are consistent with previous work (Chu et al., 2025), underscoring the importance of instruction-following capabilities in models used as starting points for RL training.

### E.3. Empirical Impact of $Q$-Value Estimation

In Section 3.3, we present the development of a practical DPSDP approach, leveraging $\pi_{\text{ref}}$ to approximate $Q$-values that are otherwise infeasible or computationally costly to obtain. We are especially interested in investigating our approximation of

| Approach | MATH 500 | | | GSM8K | | | MMLU-Pro Math | | | Olympiad Bench | | |
|---|---|---|---|---|---|---|---|---|---|---|---|---|
| | p1@t1 | m1@t5 | p1@t5 | p1@t1 | m1@t5 | p1@t5 | p1@t1 | m1@t5 | p1@t5 | p1@t1 | m1@t5 | p1@t5 |
| **Ministral-8B-It** | 55.8 | 53.4 | 58.4 | 83.4 | 81.9 | 84.7 | 52.1 | 50.6 | 55.7 | 22.8 | 22.7 | 24.8 |
| DPSDP w/o SFT | 52.6 | 53.8 | 54.4 | 90.6 | 90.8 | 90.9 | 54.1 | 54.4 | 55.5 | 26.0 | 26.1 | 26.7 |
| DPSDP (ours) | 58.2 | 63.2 | 70.0 | 87.8 | 89.1 | 92.7 | 53.1 | 54.2 | 64.3 | 25.8 | 27.0 | 32.9 |

*Table 5.* Performance comparison of DPSDP with or without preliminary training stage (SFT).

$Q_1^{\widehat{\pi}_2}(s_1, a_1)$, the $Q$-values for the feedback step. To evaluate the impact of it, we conduct experiments on a smaller dataset primarily consisting of augmented problems from MATH. We first replicate the training process from Algorithm 2 on this dataset, using $\pi_{\mathsf{ref}}$ for estimation and combining training across all turns into a single step for each agent, which we refer to as *unified training*. In contrast, we implement a step-by-step training procedure based on Algorithm 1 with a DPO loss, with each turn being trained separately. We denote this approach as *step-by-step training*. Specifically, we first use $\pi_{\mathsf{ref}}$ to sample $n$ second-turn answers, which are then used to construct a DPO dataset for actor training, yielding an intermediate actor model $\widehat{\pi}_2^a$. Next, we sample diverse feedback from the base critic and use $\widehat{\pi}_2^a$, instead of $\pi_{\mathsf{ref}}$ as in unified training, to generate refined answers based on this feedback, obtaining an accurate value of $Q_1^{\widehat{\pi}_2^a}$. We then conduct DPO training on the critic, producing a refined critic model $\widehat{\pi}_1^c$. Finally, we improve the actor's first-turn responses based on $\widehat{\pi}_2^a$, following the same approach as in Algorithm 2. To ensure broad exposure to different problems, we use three distinct problem sets at each stage of step-by-step training, each matching the size of the dataset used in unified training. We evaluate both methods on MATH 500 using Llama-based models, as shown in Table 6. Notably, we find that the policy trained via unified training outperforms the step-by-step approach, suggesting both the accuracy of our estimation strategy and the potential advantage of jointly learning answers alongside their refinements.

| Approach | MATH 500 | | |
|---|---|---|---|
| | p1@t1 | m1@t5 | p1@t5 |
| **Llama-3.1 Unified** | 54.4 | 55.2 | 59.2 |
| | | | |
| **Llama-3.1 Step-by-Step** | | | |
| + Trained on $a_2$ | 53.0 | 54.2 | 58.6 |
| + Trained on $a_1$ | 53.0 | 54.0 | 58.8 |
| + Trained on $a_0$ | 52.6 | 54.2 | 59.0 |

*Table 6.* Comparison between unified training and step-by-step training. The experiments are conducted on a smaller, MATH-dominated dataset.

### E.4. Qualitative Analysis

We present example output from DPSDP Ministral-based models[1].

---

**Critic captures errors in actor's first answer**

**Problem**   A regular hexagon can be divided into six equilateral triangles. If the perimeter of one of the triangles is 21 inches, what is the perimeter, in inches, of the regular hexagon?

**Answer 1**   To solve this problem, we need to understand the relationship between the equilateral triangles that make up the regular hexagon.

1. **Identify the Perimeter of One Triangle**: We are given that the perimeter of one equilateral triangle is 21 inches. Since all sides of an equilateral triangle are equal, each side of the triangle is $\frac{21}{3} = 7$ inches.

2. **Determine the Perimeter of the Hexagon**: A regular hexagon is made up of six equilateral triangles. There-

---

[1]All modifications to the model-generated responses presented in this section were purely formatting-related and did not affect the underlying meaning or content.

fore, the perimeter of the hexagon is the sum of the perimeters of these six triangles.

3. **Calculate the Total Perimeter**: Since each triangle has a perimeter of 21 inches, the total perimeter of the hexagon is $6 \times 21 = 126$ inches.

Thus, the perimeter of the regular hexagon is $\boxed{126}$ inches.

**Feedback 1**  Upon reviewing the previous response, I noticed a conceptual error in the calculation of the hexagon's perimeter. Specifically, the perimeter of the hexagon should not be calculated by summing the perimeters of the six triangles, as this would double-count the sides. Instead, the perimeter of the hexagon is simply the sum of the lengths of its six sides.
Here are the points to consider:

1. **Perimeter of One Triangle**: The perimeter of one equilateral triangle is given as 21 inches. Each side of the triangle is $\frac{21}{3} = 7$ inches.

2. **Perimeter of the Hexagon**: The hexagon has six sides, each of which is 7 inches long. Therefore, the perimeter of the hexagon is $6 \times 7$.

The error in the previous response was in incorrectly summing the perimeters of the triangles instead of directly calculating the perimeter from the side lengths of the hexagon.

**Answer 2**  To solve this problem correctly, we need to understand the relationship between the equilateral triangles that make up the regular hexagon and how to calculate its perimeter accurately.

1. **Identify the Perimeter of One Triangle**: We are given that the perimeter of one equilateral triangle is 21 inches. Since all sides of an equilateral triangle are equal, each side of the triangle is $\frac{21}{3} = 7$ inches.

2. **Determine the Perimeter of the Hexagon**: A regular hexagon is made up of six equilateral triangles. Each side of the hexagon is the same length as a side of these triangles.

3. **Calculate the Total Perimeter**: The perimeter of the hexagon is the sum of the lengths of its six sides. Since each side is 7 inches long, the perimeter of the hexagon is $6 \times 7 = 42$ inches.

Thus, the perimeter of the regular hexagon is $\boxed{42}$ inches.

---

## Critic iteratively helps refine the answer

**Problem**  Let $z$ be a complex number such that $z^5 = 1$ and $z \neq 1$. Compute

$$z + \frac{1}{z} + z^2 + \frac{1}{z^2}.$$

**Answer 1**  Given that $z$ is a complex number satisfying $z^5 = 1$ and $z \neq 1$, we need to compute the expression

$$z + \frac{1}{z} + z^2 + \frac{1}{z^2}.$$

First, note that since $z^5 = 1$, $z$ is a fifth root of unity. The fifth roots of unity are given by $e^{2\pi ki/5}$ for $k = 0, 1, 2, 3, 4$. Since $z \neq 1$, we have $z = e^{2\pi ki/5}$ for $k = 1, 2, 3, 4$.
Now, let's compute the expression step-by-step:
1. **Compute $z + \frac{1}{z}$:**

$$z + \frac{1}{z} = z + \frac{\overline{z}}{|z|^2} = z + \overline{z}$$

Since $z$ is a unit magnitude complex number (as it is a root of unity), $|z| = 1$, and thus $\overline{z} = \frac{1}{z}$. Therefore,

$$z + \frac{1}{z} = z + \overline{z} = z + \frac{1}{z}$$

2. **Compute $z^2 + \frac{1}{z^2}$:**

$$z^2 + \frac{1}{z^2} = z^2 + \frac{\overline{z^2}}{|z^2|^2} = z^2 + \overline{z^2}$$

Since $z^2$ is also a unit magnitude complex number, $\overline{z^2} = \frac{1}{z^2}$. Therefore,

$$z^2 + \frac{1}{z^2} = z^2 + \overline{z^2} = z^2 + \frac{1}{z^2}$$

3. **Combine the results:**

$$z + \frac{1}{z} + z^2 + \frac{1}{z^2} = (z + \frac{1}{z}) + (z^2 + \frac{1}{z^2})$$

Since $z + \frac{1}{z} = z + \overline{z}$ and $z^2 + \frac{1}{z^2} = z^2 + \overline{z^2}$, and knowing that the sum of all fifth roots of unity is zero (as they form a geometric series with sum to zero), we have:

$$z + \overline{z} + z^2 + \overline{z^2} = 0$$

Thus, the final answer is:

$$\boxed{0}$$

**Feedback 1** Upon reviewing the previous response, I noticed a few points that need attention:

1. **Sum of Fifth Roots of Unity**: The statement that the sum of all fifth roots of unity is zero is correct. However, the reasoning provided in the solution does not explicitly use this property correctly. The sum of the fifth roots of unity is indeed zero, but the solution should clearly state this and use it to simplify the expression.

2. **Simplification of the Expression**: The solution does not clearly show how the expression $z + \frac{1}{z} + z^2 + \frac{1}{z^2}$ simplifies using the properties of the fifth roots of unity. The steps should be more detailed to show the intermediate calculations.

3. **Use of Conjugates**: The solution mentions the use of conjugates but does not explicitly show how the conjugates are used in the simplification process. This could be made clearer for better understanding.

4. **Final Calculation**: The final calculation should be more explicit in showing how the sum of the terms equals zero. The solution should clearly state that the sum of the fifth roots of unity (excluding 1) is zero, and thus the expression simplifies to zero.

   By addressing these points, the solution can be made more accurate and complete.

**Answer 2** Given that $z$ is a complex number satisfying $z^5 = 1$ and $z \neq 1$, we need to compute the expression

$$z + \frac{1}{z} + z^2 + \frac{1}{z^2}.$$

First, note that since $z^5 = 1$, $z$ is a fifth root of unity. The fifth roots of unity are given by $e^{2\pi ki/5}$ for $k = 0, 1, 2, 3, 4$. Since $z \neq 1$, we have $z = e^{2\pi ki/5}$ for $k = 1, 2, 3, 4$.

Now, let's compute the expression step-by-step:

1. **Sum of Fifth Roots of Unity**: The sum of all fifth roots of unity is zero. This can be shown using the formula for the sum of a geometric series:
   $$1 + z + z^2 + z^3 + z^4 = 0$$

   Since $z \neq 1$, we exclude 1 from our consideration.

2. **Compute $z + \frac{1}{z}$**:
$$z + \frac{1}{z} = z + \overline{z}$$

Since $z$ is a unit magnitude complex number (as it is a root of unity), $|z| = 1$, and thus $\overline{z} = \frac{1}{z}$. Therefore,

$$z + \frac{1}{z} = z + \overline{z}$$

3. **Compute $z^2 + \frac{1}{z^2}$**:
$$z^2 + \frac{1}{z^2} = z^2 + \overline{z^2}$$

Since $z^2$ is also a unit magnitude complex number, $\overline{z^2} = \frac{1}{z^2}$. Therefore,

$$z^2 + \frac{1}{z^2} = z^2 + \overline{z^2}$$

4. **Combine the results**:
$$z + \frac{1}{z} + z^2 + \frac{1}{z^2} = (z + \overline{z}) + (z^2 + \overline{z^2})$$

Since the sum of all fifth roots of unity (excluding 1) is zero, we have:

$$z + \overline{z} + z^2 + \overline{z^2} = 0$$

Thus, the final answer is:

$$\boxed{0}$$

**Feedback 2** Upon reviewing the previous response, I noticed a few points that need clarification or correction:

1. **Sum of Fifth Roots of Unity**: The statement that the sum of all fifth roots of unity is zero is correct. However, the sum of the fifth roots of unity excluding 1 is not necessarily zero. The sum of the fifth roots of unity excluding 1 is actually $-1$.

2. **Compute $z + \frac{1}{z}$ and $z^2 + \frac{1}{z^2}$**: The steps to compute $z + \frac{1}{z}$ and $z^2 + \frac{1}{z^2}$ are correct, but the final combination step needs to be revisited based on the correct sum of the fifth roots of unity excluding 1.

3. **Final Combination**: The final combination step should consider the correct sum of the fifth roots of unity excluding 1, which is $-1$.

These points highlight the need to correct the final combination step to ensure the accuracy of the solution.

**Answer 3** Given that $z$ is a complex number satisfying $z^5 = 1$ and $z \neq 1$, we need to compute the expression

$$z + \frac{1}{z} + z^2 + \frac{1}{z^2}.$$

First, note that since $z^5 = 1$, $z$ is a fifth root of unity. The fifth roots of unity are given by $e^{2\pi ki/5}$ for $k = 0, 1, 2, 3, 4$. Since $z \neq 1$, we have $z = e^{2\pi ki/5}$ for $k = 1, 2, 3, 4$.
Now, let's compute the expression step-by-step:

1. **Sum of Fifth Roots of Unity**: The sum of all fifth roots of unity is zero. This can be shown using the formula for the sum of a geometric series:
$$1 + z + z^2 + z^3 + z^4 = 0$$

Since $z \neq 1$, we exclude 1 from our consideration. The sum of the other four fifth roots of unity is:

$$z + z^2 + z^3 + z^4 = -1$$

2. **Compute** $z + \frac{1}{z}$:

$$z + \frac{1}{z} = z + \overline{z}$$

Since $z$ is a unit magnitude complex number (as it is a root of unity), $|z| = 1$, and thus $\overline{z} = \frac{1}{z}$. Therefore,

$$z + \frac{1}{z} = z + \overline{z}$$

3. **Compute** $z^2 + \frac{1}{z^2}$:

$$z^2 + \frac{1}{z^2} = z^2 + \overline{z^2}$$

Since $z^2$ is also a unit magnitude complex number, $\overline{z^2} = \frac{1}{z^2}$. Therefore,

$$z^2 + \frac{1}{z^2} = z^2 + \overline{z^2}$$

4. **Combine the results**:

$$z + \frac{1}{z} + z^2 + \frac{1}{z^2} = (z + \overline{z}) + (z^2 + \overline{z^2})$$

We know that $z + z^2 + z^3 + z^4 = -1$. Since $z + \overline{z}$ and $z^2 + \overline{z^2}$ are real numbers (as they are sums of complex conjugates), we can write:

$$z + \overline{z} + z^2 + \overline{z^2} = -1$$

Thus, the final answer is:

$$\boxed{-1}$$

---

## Failure Pattern: Over-thinking on easy tasks

**Problem**   Marissa is hiking a 12-mile trail. She took 1 hour to walk the first 4 miles, then another hour to walk the next two miles. If she wants her average speed to be 4 miles per hour, what speed (in miles per hour) does she need to walk the remaining distance?

**Answer 1**   To solve this problem, we need to calculate the total time Marissa has spent hiking and the distance she has covered so far. Then, we can determine the speed she needs to maintain for the remaining distance to achieve an average speed of 4 miles per hour.

1. **Calculate the distance covered so far:** Marissa has walked the first 4 miles in 1 hour and the next 2 miles in 1 hour. So, the total distance covered so far is:

$$4 \text{ miles} + 2 \text{ miles} = 6 \text{ miles}$$

2. **Calculate the time spent so far:** Marissa has spent 1 hour walking the first 4 miles and another hour walking the next 2 miles. So, the total time spent so far is:

$$1 \text{ hour} + 1 \text{ hour} = 2 \text{ hours}$$

3. **Determine the total time required to complete the trail at an average speed of 4 miles per hour:** The total distance of the trail is 12 miles. To maintain an average speed of 4 miles per hour, the total time required to complete the trail is:

$$\frac{12 \text{ miles}}{4 \text{ miles per hour}} = 3 \text{ hours}$$

4. **Calculate the remaining time to complete the trail:** Marissa has already spent 2 hours hiking. Therefore, the remaining time to complete the trail is:

$$3 \text{ hours} - 2 \text{ hours} = 1 \text{ hour}$$

5. **Calculate the remaining distance:** The total distance of the trail is 12 miles, and Marissa has already covered 6 miles. Therefore, the remaining distance is:

$$12 \text{ miles} - 6 \text{ miles} = 6 \text{ miles}$$

6. **Determine the speed required to cover the remaining distance in the remaining time:** To cover the remaining 6 miles in 1 hour, Marissa needs to walk at a speed of:

$$\frac{6 \text{ miles}}{1 \text{ hour}} = 6 \text{ miles per hour}$$

Therefore, Marissa needs to walk at a speed of $\boxed{6}$ miles per hour to cover the remaining distance and maintain an average speed of 4 miles per hour for the entire trail.

**Feedback 1**  Upon reviewing the previous response, I noticed a few points that need attention:

1. **Average Speed Calculation**: The average speed for the entire trail should be calculated based on the total distance and the total time. The initial calculation of the total time required to complete the trail at an average speed of 4 miles per hour is correct.

2. **Remaining Distance and Time**: The calculation of the remaining distance and the remaining time is correct. However, the speed required to cover the remaining distance in the remaining time should be re-evaluated to ensure it aligns with the average speed requirement.

3. **Speed Requirement**: The speed required to cover the remaining distance in the remaining time should be recalculated to ensure it meets the average speed requirement for the entire trail.

These points highlight areas where the solution might need further verification or adjustment to ensure accuracy and completeness.

**Answer 2**  Upon reflecting on the previous response, let's correct and clarify the steps to ensure accuracy.

1. **Calculate the distance covered so far:** Marissa has walked the first 4 miles in 1 hour and the next 2 miles in 1 hour. So, the total distance covered so far is:

$$4 \text{ miles} + 2 \text{ miles} = 6 \text{ miles}$$

2. **Calculate the time spent so far:** Marissa has spent 1 hour walking the first 4 miles and another hour walking the next 2 miles. So, the total time spent so far is:

$$1 \text{ hour} + 1 \text{ hour} = 2 \text{ hours}$$

3. **Determine the total time required to complete the trail at an average speed of 4 miles per hour:** The total distance of the trail is 12 miles. To maintain an average speed of 4 miles per hour, the total time required to complete the trail is:
$$\frac{12 \text{ miles}}{4 \text{ miles per hour}} = 3 \text{ hours}$$

4. **Calculate the remaining time to complete the trail:** Marissa has already spent 2 hours hiking. Therefore, the remaining time to complete the trail is:

$$3 \text{ hours} - 2 \text{ hours} = 1 \text{ hour}$$

5. **Calculate the remaining distance:** The total distance of the trail is 12 miles, and Marissa has already covered 6 miles. Therefore, the remaining distance is:

$$12 \text{ miles} - 6 \text{ miles} = 6 \text{ miles}$$

6. **Determine the speed required to cover the remaining distance in the remaining time:** To cover the remaining 6 miles in 1 hour, Marissa needs to walk at a speed of:

$$\frac{6 \text{ miles}}{1 \text{ hour}} = 6 \text{ miles per hour}$$

However, we need to ensure that this speed aligns with the average speed requirement for the entire trail. The average speed is calculated as the total distance divided by the total time. Marissa has already walked 6 miles in 2 hours, which is an average speed of:

$$\frac{6 \text{ miles}}{2 \text{ hours}} = 3 \text{ miles per hour}$$

To maintain an average speed of 4 miles per hour for the entire 12-mile trail, she needs to cover the remaining 6 miles in:

$$\frac{6 \text{ miles}}{4 \text{ miles per hour}} = 1.5 \text{ hours}$$

This means Marissa needs to adjust her pace to cover the remaining 6 miles in 1.5 hours, which is a speed of:

$$\frac{6 \text{ miles}}{1.5 \text{ hours}} = 4 \text{ miles per hour}$$

Therefore, Marissa needs to walk at a speed of $\boxed{4}$ miles per hour to cover the remaining distance and maintain an average speed of 4 miles per hour for the entire trail.

---

## Failure Pattern: Over-pessimistic feedback

**Problem** The results of a cross-country team's training run are graphed below. Which student has the greatest average speed?

```
[asy]
for ( int i = 1; i <= 7; ++i )
{

draw((i,0)--(i,6));
}

for ( int i = 1; i <= 5; ++i )
{

draw((0,i)--(8,i));
}
draw((-0.5,0)--(8,0), linewidth(1));
draw((0,-0.5)--(0,6), linewidth(1));
label("$O$", (0,0), SW);
label(scale(.85)*rotate(90)*"distance", (0, 3), W);
label(scale(.85)*"time", (4, 0), S);
dot((1.25, 4.5));
label(scale(.85)*"Evelyn", (1.25, 4.8), N);
```

```
dot((2.5, 2.2));
label(scale(.85)*"Briana", (2.5, 2.2), S);
dot((4.25,5.2));
label(scale(.85)*"Carla", (4.25, 5.2), SE);
dot((5.6, 2.8));
label(scale(.85)*"Debra", (5.6, 2.8), N);
dot((6.8, 1.4));
label(scale(.85)*"Angela", (6.8, 1.4), E);
[/asy]
```

**Answer 1**   [*Resoning...*] Comparing the average speeds, we find that Evelyn has the greatest average speed.

$$\boxed{Evelyn}$$

**Feedback 1**   Upon reviewing the previous response, I noticed a few errors and gaps: [*Feedback...*]
**Answer 2**   [*Resoning...*] Comparing the average speeds, we find that Carla has the greatest average speed.

$$\boxed{Carla}$$

**Feedback 2**   Upon reviewing the previous response, I noticed a few errors and gaps: [*Feedback...*]
**Answer 3**   [*Resoning...*] Comparing the average speeds, we find that Debra has the greatest average speed.

$$\boxed{Debra}$$

**Feedback 3**   Upon reviewing the previous response, I noticed a few errors and gaps: [*Feedback...*]
**Answer 4**   [*Resoning...*] Comparing the average speeds, we find that Angela has the greatest average speed.

$$\boxed{Angela}$$

