# OpenReview forum: "Reinforce LLM Reasoning through Multi-Agent Reflection"
_ICML.cc/2025/Conference — ICML 2025 poster_

### Official Review · Reviewer_X1o3 · 2025-03-07

**Overall Recommendation:** 2

**Summary:**

This paper introduces DPSDP (Direct Policy Search by Dynamic Programming), a algorithm designed to enhance the reasoning abilities of large language models by utilizing a multi-agent system. This paper concludes that DPSDP provides a robust solution for refining reasoning tasks in LLMs, allowing them to generate more accurate responses through iterative refinement and effective collaboration between multiple agents.

## update after rebuttal
Given the rebuttal and discussions with the authors, my main concern of the fair comparison in the experiments is not fully addressed. I still think this is a borderline work, so I will keep my score.

**Claims And Evidence:**

Yes

**Essential References Not Discussed:**

No.

**Experimental Designs Or Analyses:**

Yes, I checked the soundness and validity of several experimental designs and analyses, focusing primarily on those presented in Sections 4.2 (Main Results) and 4.3 (Ablation Study), as well as some from Appendix E.  The experimental designs are generally sound and well-justified, with thorough ablations and relevant comparisons. The issues are primarily areas for potential more detailed analysis, rather than fundamental flaws. The paper provides good evidence for the effectiveness of DPSDP, given the constraints of the chosen problem domain.

**Methods And Evaluation Criteria:**

Yes

**Other Comments Or Suggestions:**

No.

**Other Strengths And Weaknesses:**

## Strengths：
1. This paper establishes a strong theoretical foundation for DPSDP, providing a formal proof that, under specific conditions, the algorithm's performance can match any comparator policy within the training distribution. The proof is presented rigorously and appears sound.
2. The experimental evaluation is adequate, utilizing multiple metrics (pass1@turn1, maj1@turn5, pass1@turn5) across two base models. This provides a comprehensive assessment of DPSDP's performance. The ablation studies in the Appendix further validate the effectiveness of key design choices.
3. The paper demonstrates DPSDP's ability to improve LLM performance not only on in-distribution benchmarks but also on OOD data. This highlights the algorithm's generalization capabilities, particularly on challenging reasoning problems.

## Weaknesses：
1. While framed within a reinforcement learning context, DPSDP's two-stage training process (SFT + DPO) more closely resembles a supervised learning approach with refined training data. Crucially, the optimized model cannot be used for on/off-policy data generation and further iterative improvement, a hallmark of many RL algorithms.
2. Although the paper compares DPSDP against several baselines, the comparison to recent SOTA methods is limited. While Appendix A mentions related works, a direct comparison with a contemporary, high-performing approach is missing.
3. The paper demonstrates DPSDP's superiority over SFT. However, the SFT baseline is trained for only one epoch, likely preventing it from reaching convergence. DPSDP, in contrast, undergoes more extensive training. This discrepancy in training duration makes the comparison between DPSDP and SFT potentially unfair, as the observed performance gains might be attributed to the difference in training extent rather than the inherent advantages of the DPSDP strategy.
4. From Table 1 to Table 4, I noticed that the authors did not conduct self-critique experiments for comparison. Given that the open-source community has recently demonstrated the significant effectiveness of self-criticism and reflection capabilities in single-agent systems, I suggest the authors consider including such experiments in their comparisons. While the multi-agent approach shows advantages, the potential of single-agent methods should not be overlooked.
5. Regarding the method described in Sec 3.3, where Monte Carlo sampling of $a_2$'s accuracy is used to estimate the expected return of $a_1$, I have the following concerns: This estimation method may lead to an underestimation of $a_1$ Q-value. More critically, it might induce model hallucinations where $a_1$ generates meaningless content, yet $a_2$ still provides correct answers. I recommend the authors thoroughly investigate this potential issue and discuss possible solutions.
6. In terms of optimization algorithm selection, while the introduction of DPO is reasonable, it lacks novelty. To enhance the soundness of using DPO as an optimization method, I suggest the authors include comparisons with classic reinforcement learning baseline methods in Table 1, such as reject sampling with finetuning, and Independent PPO. This would not only highlight the advantages of DPO but also provide readers with a more comprehensive evaluation of methods.

**Questions For Authors:**

1. The paper highlights the use of DPO as a key component of DPSDP. To rigorously demonstrate the advantage of DPO over SFT, could the authors provide experimental results comparing models trained with DPO and SFT under identical conditions? Specifically, this comparison should control for either the number of training steps or the amount of training data used, ensuring a fair evaluation of the two training paradigms. This would help isolate the specific contribution of DPO.

2. The results in Table 1 show that the single-agent approach outperforms the multi-agent DPSDP on several metrics using the Mistral-8B-It base model. To further validate the effectiveness of the multi-agent design in DPSDP, could the authors provide corresponding results for the single-agent approach using the Llama-3.1-8B-It base model? This would allow for a more consistent comparison across different base models and strengthen the claims regarding the benefits of the multi-agent setup.

3. Section 3.4 indicates that the initial SFT phase utilizes high-quality feedback and refined answers produced by a more capable model. Is DPSDP's performance necessarily dependent upon this reliance on data from a stronger model? If not necessarily, it could offer valuable insights into the algorithm's self-improvement capabilities and lessen its dependence on external resources.

4. Regarding the extension of MDP policy in multi-round critique models, it is recommended that the authors conduct the following additional analytical experiments to explore in-depth the impact of multi-round critiques on model performance:
a) Plot the performance curve of the model as the number of critique rounds increases. This will help understand the evolution trend of model performance, including whether there are performance peaks or saturation points.
b) Evaluate the impact of multi-round critiques on the quality of generated content. Pay particular attention to whether there are issues such as quality degradation, content repetition, or inconsistency.
c) Under the same computational resource constraints, compare the performance differences between the baseline method (using self-consistency answer filtering) and the multi-round critique method.
This will help to more fairly assess the effectiveness of the MDP strategy.
It is suggested to present the results of these analytical experiments separately from Table 1 to provide a more comprehensive performance evaluation.

5. Regarding the data filtering process for initializing actor and critic models, it is recommended to provide detailed explanations of the data filtering process in Appendix D.1, including but not limited to the following aspects:
a) Specific steps and criteria for data cleaning (such as removing duplicates, low-quality, or irrelevant data), and any specific language models or algorithms used to assist in the filtering process.
b) How to ensure that the filtered dataset is representative and diverse for initializing actor and critic models.
Providing these details will help readers better understand the experimental setup and improve the reproducibility of the research.

**Relation To Broader Scientific Literature:**

The paper's key contributions relate to several strands of existing scientific literature, building upon and extending prior work in specific ways:

1. Intrinsic and External Self-Correction: Previous research explores both intrinsic self-correction (LLMs refining outputs without external help) and self-correction with external feedback. Intrinsic methods often involve prompting LLMs to reflect and revise, but some unrealistically assume access to correct answers. Others try training models for self-correction, finding supervised fine-tuning insufficient and exploring RL approaches, sometimes focusing on single-turn refinement. Multi-turn refinement has also been explored. However, LLMs often struggle with purely intrinsic self-correction. Research with external feedback frequently uses code generation scenarios with feedback from tests or compilers, or incorporates external tools. Some use feedback from other models, but typically treat the answer generator and feedback provider as separate entities, relying on fixed feedback or training a separate corrector. DPSDP addresses multi-turn refinement, but with a different approach and a theoretical guarantee. The LLM critic provides a flexible feedback space, unlike the restricted feedback in some previous work.

2. Multi-Agent Systems in LLMs:  The paper builds on the growing interest in multi-agent LLM systems (Guo et al., 2024b; Motwani et al., 2024). It cites examples of both competitive (debate-style) and cooperative multi-agent systems. It specifically mentions works that use multi-agent systems for reasoning improvement. DPSDP is a multi-agent system with actor and critic. It differs from some prior work by having a joint training process using DPO. It also contrasts with Motwani et al. (2024), which uses a three-model system and focuses on a 2-turn refinement, by providing a theoretical guarantee and enabling multi-turn refinement.

**Theoretical Claims:**

Yes. The proof of Theorem 1 is reasonable in its approach, but the assumptions it relies upon may not hold in the context of LLMs, and some of the relaxations used in the proof may be too loose. Therefore, the theoretical guarantee provided by Theorem 1 may not be strong. More detailed analysis and experiments are needed to verify the actual performance of the DPSDP algorithm.

---

> ### Author Rebuttal · Authors · 2025-04-01
>
> Thanks for reviewing our paper!
>
> # Weaknesses
>
> **Q: DPSDP does not resemble an RL algorithm**
>
> We adapt our algorithm from PSDP, a classic reinforcement learning method, and formulate iterative refinement as a standard MDP (Section 2). As shown in Algorithm 1, we optimize each step in reverse via policy rollout and update, following the PSDP framework.
>
> A key insight is our estimation of ground-truth Q-values, which removes inter-iteration dependencies and simplifies implementation. Experiments validate this approach, and we compare the theoretical and practical versions in Appendix E.5.
>
> **Q: Other SOTA models baselines**
>
> While many works aim to improve LLM responses, few share our settings. For instance, SCoRe targets one-shot self-improvement without external feedback. The closest is RISE with an oracle, which assumes access to ground-truth correctness at test-time.
>
> We include this variant as a baseline, re-implementing it by first training an SFTed Ministral-8B-It model (Section 3.4), followed by reinforcement learning on the same problem set used for our main models.
>
> Task|Model|pass@1|maj@t3|maj@t5|pass@t5
> -|-|-|-|-|-
> MATH|DPSDP|58.2|61.8|63.2|70.0
> ||Oracle-RISE|59.2|64.2|65.4|65.8
> GSM8K|DPSDP|87.8|89.1|89.1|92.7
> ||Oracle-RISE|88.9|92.1|92.6|92.9
> MMLU Pro Math|DPSDP|53.1|53.0|54.2|64.3
> ||Oracle-RISE|52.8|59.8|61.3|62.4
> Olympiad-Bench|DPSDP|25.8|27.2|27.0|32.9
> ||Oracle-RISE|25.8|30.3|30.6|30.9
>
> Our results show that our models achieve maj@t5 accuracies comparable to RISE on challenging benchmarks such as MATH and Olympiad. Our models consistently outperform RISE on pass@t5, indicating that the actor—guided by critic feedback—explores the solution space more actively rather than sticking to initial responses.
>
>
> **Q: Unfair comparison with SFT**
>
> It is worthnoting that the rows labeled +SFT in Table 1 are not SFT baselines but intermediate results from preliminary training (Section 3.4), which our DPSDP models build upon. For a fair comparison with standard SFT, we evaluated against STaR baselines (Section 4.2), which used the same SFTed base models, problem set, number of trajectories, and training epochs. Results show that STaR fails to enable self-improvement, underscoring the effectiveness of our approach.
>
> **Q: Comparison with self-critique**
>
> To highlight the benefits of the multi-agent setup, we replicated the full training—preliminary training and DPSDP—using a single model as both actor and critic. As shown in Table 1 (Single-Agent row) and discussed in Section 4.3, this setup consistently underperforms the multi-agent system, especially on harder benchmarks. We confirmed this across LLaMA-based models (see reply to Reviewer jmss, Table 1), and our conclusion holds.
>
> **Q: Concerns with Q-value estimation with Monte Carlo**
>
> Q-values reflect expected cumulative rewards rather than the immediate correctness of individual actions like feedback $a_1$. The correctness of $a_2 \sim \pi(\cdot \mid s_2)$ provides an unbiased estimate of the Q-value. A related example is DeepSeek-R1-Zero, which shows reduced readability during reasoning but achieves strong final performance.
>
> **Q: The soundness of using DPO as optimization method**
>
> Our core contribution is introducing PSDP for multi-agent LLM refinement. While we use DPO for optimization, it’s a modular component that can be replaced (e.g., with rejection sampling or KTO) without altering the overall approach.
>
> # Questions For Authors
>
> **Q: Fair comparison with SFT baseline**
>
> Please see the response above.
>
> **Q: Single-Agent experiment with Llama-based models**
>
> We present single-agent results using Llama-based in reply to Reviewer jmss, Table 1, and our findings consistently demonstrate the general superiority of the multi-agent system over the single-agent setup.
>
> **Q: Necessity of Preliminary Training (SFT)**
>
> See the reply to Reviewer Hay1, Table 2.
>
> **Q: Additional analytical experiments to explore impact of multi-round critiques**
>
> 1. We visualize the accuracy dynamics in Figure 3 and provide results for more refinement steps in reply to Reviewer jmss, Table 3. Further detailed metrics are also presented in reply to Reviewer 2FHw, Table 1.
>
> 1. In reply to Reviewer 2FHw, we identify several failure patterns and illustrate how iterative refinement helps mitigate issues related to over-refinement.
>
> 1. In Appendix E.1, we compare maj1@t5 and maj5@t1, showing that the performance gains stem from the refinement process itself rather than from a simple increase in test-time computation.
>
> **Q: Data processing**
>
> Our data filtering is intentionally simple and transparent to attribute performance gains to our algorithm rather than data curation. We remove duplicates and randomly sample problems for unbiased coverage. For the SFT dataset, we use an oracle model to generate refined answers and filter feedback (Section 3.4). To ensure diverse reasoning styles, we include trajectories from different model famalies as explained in Appendix D.1.

---

> > ### Comment · Reviewer_X1o3 · 2025-04-07
> >
> > Thanks for the author's feedback. It partially addressed my concerns, but not all my questions and weaknesses are fully explained. For example, the response to the comparison with SFT seems to avoid directly answering the question. I understand your model is built upon an intermediate SFT ckpt, what I mean is the `+SFT` can be still fully trained it until convergence, then compared with the proposed approach. This is a fair comparison with SFT.
> >
> > In general, I still think this is still a borderline work.

---

> > > ### Author Response · Authors · 2025-04-08
> > >
> > > We thank the reviewer for their insightful comments and the suggestion to compare DPSDP against a well-trained SFT algorithm. However, we would like to respectfully highlight a crucial difference in the learning paradigms between well-trained SFT and our proposed algorithm (DPSDP):
> > > - In the well-trained SFT, the model learns directly from the high-quality **oracle/expert data** (see, e.g., Appendix D.1) via a behavior cloning objective.
> > > - In contrast, DPSDP is designed to avoid the requirement of high-quality oracle/expert data. It only uses preference pairs derived from **self-generated data** and rule-based correctness evaluations (Sec 3.3), without further direct access to the oracle’s outputs during this core optimization stage.
> > >
> > > *Therefore, DPSDP and the well-trained SFT are not directly comparable, because a well-trained SFT distills from high-quality expert data, whereas DPSDP only learns and refines from its own.* This constitutes an unfair comparison for DPSDP (as well-trained SFT could access additional information) and cannot directly evaluate the effectiveness of the self-improvement mechanism in DPSDP.
> > >
> > > On the other hand, we believe our comparison against STaR, which also learns from self-generated data via an SFT-like objective, provides a fair comparison of different methods (DPSDP vs. SFT based refinement) designed to enable iterative refinement based on the agents’ own experience.
> > >
> > > We hope this helps address the concerns, and we would be happy to discuss any further questions.

---

### Official Review · Reviewer_2FHw · 2025-03-10

**Overall Recommendation:** 3

**Summary:**

This paper proposes a new reinforcement learning algorithm, DPSDP, to enhance the mathematical reasoning capabilities of large language models using a multi-agent approach involving an actor and a critic. The method instantiates two LLMs as actors and critics to perform self-reflection-style reasoning, collecting preference data by sampling from the models. This process consists of at most two rounds: posing a question, generating an initial response, providing feedback, and generating a revised response. The preference is estimated by rolling out the policy.

The DPSDP algorithm is evaluated on four benchmarks, covering both in-distribution and out-of-distribution settings, using two base models: Mistral-8B-It and Llama-3.1-8B-It. The results show improvements in accuracy across benchmarks and settings, outperforming baselines such as STaR and STaR-DPO. The authors also conduct an ablation study to analyze the impact of single-agent versus multi-agent approaches, Markovian versus non-Markovian formulations, and generative versus non-generative critics.

## Response
I maintain my view that this paper is suitable for acceptance.

**Claims And Evidence:**

In general, I think the theoretical analysis is not highly relevant to the practical algorithm because extensive modifications are made to ensure feasibility. Aside from this point, the other claims are reasonable.

**Essential References Not Discussed:**

N/A

**Experimental Designs Or Analyses:**

My main concern is the experimental section. The study primarily focuses on Ministerial-8B-It, with only a few experiments on Llama-3.1-8B-It. As a result, the experiments for Llama-3.1-8B-It seem incomplete. Additionally, Ministerial-8B-It is not a strong mathematical model. Many models, such as Qwen , perform better, and there are also models fine-tuned on mathematical datasets like deepseek-math and Qwen-math. These models would provide a more reasonable baseline for improvement. Furthermore, some papers (e.g., https://arxiv.org/abs/2310.01798) suggest that large language models cannot self-improve. An ablation study should be conducted to address this concern. (I am also unsure how supervised fine-tuning (SFT) was tested in the experiments—perhaps this concern has already been addressed.)

**Methods And Evaluation Criteria:**

This method makes sense to me. However, I think the evaluation metrics need more validation. Why are m1@p5 and p1@t5 appropriate metrics? Have you considered using 10 instead? Additionally, for the accuracy of m1@p5 and p1@t5, could you analyze the failure patterns in more detail? I would appreciate the metrics and analysis being as well-explained as in the paper "Self-Rewarding Correction for Mathematical Reasoning".

**Other Comments Or Suggestions:**

If the author could address my concern about the experiment part, I would be happy to increase the score.

**Other Strengths And Weaknesses:**

N/A

**Questions For Authors:**

I am a bit unclear about the difference between Markovian and non-Markovian cases. What is the context for each?

**Relation To Broader Scientific Literature:**

This contribute to the self-improvement and self-correction topic of LLM. It also contribute a new preference training method.

**Theoretical Claims:**

I didn't check the proofs.

---

> ### Author Rebuttal · Authors · 2025-04-01
>
> Thanks for reviewing our paper!
>
> # Claims And Evidence
>
> **Q: Analysis is not highly related to the practical algorithm**
>
> We provide further analysis on how approximation in the practical algorithm affects the theoretical results in reply to Reviewer Hay1.
>
>
> **Q: More detailed metrics and failure pattern analysis**
>
> We adopt the metrics p1@t1, m1@t5, and p1@t5 in line with prior work [1], and provide additional evaluation details for a more comprehensive analysis.
>
> First, we scale up the number of test-time refinement iterations, as shown in reply to Reviewer jmss, Table 3.
>
> Next, we analyze the dynamics of accuracy over the course of refinement. Using models based on Ministral-8B-Instruct as a representative example, we plot the changes in accuracy across iterations in Figure 3. For each refinement step, we define $\Delta^{c \rightarrow i}$ as the proportion of problems that change from correct to incorrect after refinement, and $\Delta^{i \rightarrow c}$ as the proportion that transition from incorrect to correct.
>
> |Iteration|$\Delta^{i\rightarrow c}$|$\Delta^{c\rightarrow i}$|
> |-|-|-|
> |t1→t2|7.8|4.0|
> |t2→t3|4.0|3.6|
> |t3→t4|3.4|2.8|
> |t4→t5|3.0|2.8|
> |t5→t6|1.8|1.6|
> |t6→t7|2.0|2.4|
> |t7→t8|1.4|1.0|
>
> The table illustrates two key observations:
>
> 1. $\Delta^{i \rightarrow c}$ consistently exceeds $\Delta^{c \rightarrow i}$, indicating that the refinement process is generally beneficial.
>
> 2. Both $\Delta^{i \rightarrow c}$ and $\Delta^{c \rightarrow i}$ decrease as the number of iterations increases, suggesting an initial exploratory phase followed by stabilization in later refinement steps.
>
> We conducted the same analysis on models based on Llama and Qwen and observed similar patterns. Due to space constraints, we omit those results here.
>
> In addition to the qualitative analysis presented in Appendix E.6, we identified several notable failure patterns:
>
> - Answer enumeration: The critic repeatedly provides negative feedback, prompting the actor to cycle through different answers at each turn—effectively enumerating possible solutions.
>
> - Answer degradation: The critic incorrectly assigns negative feedback, leading the actor to progressively degrade a previously correct answer. However, this over-refinement issue is relatively rare (as evidenced by small $\Delta^{c \rightarrow i}$) and can be mitigated by later refinement steps.
>
> - Incorrect feedback tolerance: Occasionally, a correct answer is incorrectly revised due to faulty feedback. Yet, subsequent iterations can recover the correct answer and lead to correct final answer by majority voting across all turns, helping to mitigate the effects of over-refinement.
>
> # Experimental Designs Or Analyses
>
> **Q: diverse base models**
>
> We conducted DPSDP on Qwen2.5-3B and the results are presented in reply to Reviewer Hay1, Table 1.
>
> **Q: Investigate whether models can self-improve**
>
> While earlier work suggested that models are unable to self-improve, recent studies—such as SCoRe and RISE—have demonstrated that large language models (LLMs) can develop self-improvement capabilities when properly trained. To further explore this, we conducted an ablation study in which a single model served as both the actor and critic. The results are shown in Table 1 under the row labeled Single-Agent. A detailed comparison between the single-agent and multi-agent setups is provided in Section 4.3, under the paragraph titled *Single-Agent vs. Multi-Agent*.
>
> Our findings are consistent with those of SCoRe and RISE: a single model can indeed self-improve. However, its performance is generally weaker than that of the multi-agent system, particularly on more challenging benchmarks. We replicated the single-agent setup using Llama-based models and presented the results in reply to Reviewer jmss, Table 1. Our conclusion remains the same.
>
>
>
> **Q: SFT baseline**
>
> One of our baselines, STaR, serves as the SFT counterpart to our algorithm. To ensure a fair comparison, STaR was implemented using the same SFTed models and trained on the identical prompt set. However, as discussed in Section 4.2, STaR fails to enable effective self-improvement in models.
>
> # Questions For Authors
>
> **Q: Difference between Markovian and non-Markovian**
>
> In Section 3.3, under the paragraph titled Optimizing Iterative Refinement with Reduced Complexity, we define the transition function $\delta(s_h, a_h)$, which reflects a Markovian setting. This design includes only the most recent answer and feedback in the prompt, removing all prior conversational history. The motivation behind this choice is the heuristic that recent context is more informative and relevant than earlier interactions.
>
> In contrast, an alternative approach includes the entire conversation history—i.e., all previous responses and feedback—in the prompt. This setting diverges from the Markov Decision Process (MDP) we defined and is therefore referred to as non-Markovian.
>
> [1] Recursive Introspection: Teaching Language Model Agents How to Self-Improve

---

### Official Review · Reviewer_jmss · 2025-03-13

**Overall Recommendation:** 3

**Summary:**

- The focus of the paper is on verification and refinement with an actor and critic model, using a method that trains on self-generated data
	- the actor model generates and refines responses based on feedback from a critic
	- the actor and critic are jointly trained with RL
- The authors propose a dynamic programming-based approach to optimizing the policies of the actor and critic jointly
	- an analysis of the approach is given in Theorem 1
- The approach is evaluated in practice on two LLMs (Ministral-8B-Instruct and Llama-3.1-8B-Instruct), which are first SFT'd on feedback data and then finetuned wtih DPSDP.
	- evaluation is done on four math datasets, two of which are out of domain
	- the method is compared against STaR and STaR-DPO in some of the settings
- On Ministral-8B, DPSDP outperforms STaR-DPO on MATH 500 and MMLU-Pro MATH. It performs comparably on GSM8K and Olympiad Bench.

## update after rebuttal
The rebuttal by the authors has helped address several of my larger concerns and I have chosen to increase my score to a 3.

**Claims And Evidence:**

- The authors claim the method outperforms baselines across datasets. This is only partially supported, as the baseline is only implemented for one of the two models.

**Essential References Not Discussed:**

NA

**Experimental Designs Or Analyses:**

I am concerned by the fact that the baseline was only implemented on one model. What was the reason for omitting the STaR and STaR-DPO baseline for Llama3.1 8B?

**Methods And Evaluation Criteria:**

The method is not clear, and requires a bit more intuition. Specifically, the explanation in L153-159 (right) does not make it clear how the pairwise data is collected and verified, i.e. how are $a^1$ and $a^2$ determined? According to Algorithm 1, they are both sampled from $\pi_{\text{ref}}$ but it's not clear how one is marked as preferred and the other as dispreferred.

Notational issues:
- as far as I can tell, $d_0^\pi$ and $d_h^{\pi_{\text{ref}}}$ are not defined (line 126, equation (1)). I assume these are trajectories?

**Other Comments Or Suggestions:**

- I think there's some kind of spacing issue with L264-274
- Overall the spacing/presentation of the paper could be refined.

Typos:
- L154 (right): cross-entropy
- L156 (right): a collected pairwise dataset.
- L308: challenging
- L314: Olympiad-level

**Other Strengths And Weaknesses:**

Jointly training the actor and critic for reasoning refinement is an interesting direction and seems to be novel.

**Questions For Authors:**

What is the stopping criterion of the refinement process? With refinement, there is often a problem of over-refinement, where a correct answer is refined and turned incorrect. Does training the models to perform iterative refinement address this? At test time, do models know when to stop/have you measured overrefinement?

**Relation To Broader Scientific Literature:**

The related work section is in the appendix but is fairly complete, covering most relevant work.

**Theoretical Claims:**

Based on Assumptions 1 and 2, the authors claim that the policy resulting from their method competes with ``any policy under single-policy concentrability and bounded in-distribution generalization error.''.

To me, this theoretical claim does not add much as-is; the success of this kind of paper rests on its results in practice.

---

> ### Author Rebuttal · Authors · 2025-04-01
>
> Thanks for the efforts in reviewing our paper! We will take your suggestions, fix the typo and revise the presentations accordingly in the next revision!
>
> # Methods And Evaluation Criteria
>
> **Q: unclear how $a_1$ and $a_2$ are labeled as chosen and rejected actions**
>
> Algorithm 1 presents the theoretical version of our method and does not explicitly label $a_1$ and $a_2$ as chosen or rejected. Instead, it assumes access to the Q-value function, and the Q-values of $a_1$ and $a_2$ are directly used in the cross-entropy loss (see line 176, left column).
>
> In the practical implementation (Algorithm 2), we estimate the Q-values as described in Section 3.3, under *Estimation of Q-values*. Specifically, we approximate the Q-values based on the correctness of responses. For each action pair $(a_1, a_2)$, the action with the higher estimated Q-value is labeled as "chosen," and the other as "rejected." We justify the reliability of this estimation both intuitively (Section 3.3) and empirically (Section 4 and Appendix E.5).
>
> **Q: Notation issues -- definition of $d_h^{\pi}$**
>
> We adopt standard notation from the reinforcement learning literature, where $d_h^{\pi}$ denotes the distribution over the state space at step $h$ when following policy $\pi$ (see lines 97–98, right column). Specifically, when $h = 0$, $d_0^{\pi}$ represents the initial state distribution—i.e., the distribution over prompts drawn from the prompt set or provided by users. When $\pi = \pi_\mathsf{ref}$, $d_h^{\pi_\mathsf{ref}}$ refers to the state distribution at step $h$ under the reference policy $\pi_\mathsf{ref}$.
>
> # Experimental Designs Or Analyses
>
> **Q: Baseline implementations with Llama-based models**
>
> To further validate the effectiveness of our algorithm, we benchmarked it against STaR and STaR-DPO using two additional base models: Llama 3.1-8B-Instruct and Qwen 2.5-3B. We also replicated the single-agent setup using Llama-based models as discussed in Section 4.3. Across both settings, our algorithm consistently outperforms the baselines, demonstrating robust and superior performance.
>
> - Llama
>     |Task|Model|pass@t1|maj@t5|pass@t5|
>     |-|-|-|-|-|
>     |MATH|DPSDP(Llama)|55.8|58.4|62.0|
>     ||STaR|50.8|52.2|56.8|
>     ||STaR-DPO|54.2|55.6|59.2|
>     ||Single-Agent|53.4|54.8|58.0|
>     |GSM8K|DPSDP(Llama)|87.5|88.4|91.2|
>     ||STaR|83.6|81.3|87.5|
>     ||STaR-DPO|87.5|87.4|90.3|
>     ||Single-Agent|87.9|87.6|90.4|
>     |MMLU-ProMath|DPSDP(Llama)|56.6|58.0|62.1|
>     ||STaR|53.8|54.6|58.5|
>     ||STaR-DPO|54.8|55.0|60.3|
>     ||Single-Agent|56.1|57.3|62.0|
>     |OlympiadBench|DPSDP(Llama)|22.4|23.0|25.1|
>     ||STaR|20.5|20.3|22.4|
>     ||STaR-DPO|20.9|21.5|24.3|
>     ||Single-Agent|23.0|21.5|25.1|
>
> - Qwen
>     |Task|Model|pass@t1|maj@t5|pass@t5|
>     |-|-|-|-|-|
>     |MATH|DPSDP(Qwen)|60.4|62.0|65.2|
>     ||STaR|59.0|59.6|64.8|
>     ||STaR-DPO|60.4|60.2|64.8|
>     |GSM8K|DPSDP(Qwen)|79.9|79.9|84.2|
>     ||STaR|80.3|79.5|83.7|
>     ||STaR-DPO|79.4|78.9|82.6|
>     |MMLU-Pro Math|DPSDP(Qwen)|52.6|53.2|57.1|
>     ||STaR|51.9|51.8|56.8|
>     ||STaR-DPO|51.2|52.3|55.9|
>     |OlympiadBench|DPSDP(Qwen)|24.0|24.0|26.0|
>     ||STaR|23.3|22.6|24.8|
>     ||STaR-DPO|23.1|22.8|28.9|
>
> # Questions For Authors
>
> **Q: Stopping Criterion and Overrefinement**
>
> We did not implement a specific stopping criterion for the refinement process, aside from a fixed limit on the number of refinement iterations, set to 5 in our experiments. To study the potential issue of over-refinement, we extended the number of iterations to 11. Our results show that accuracy generally improves with more refinement steps—reflecting the benefit of increased test-time computation—until it plateaus around 5 to 7 iterations. Beyond that, performance remains stable and shows minimal degradation, suggesting over-refinement is not a significant concern in practice.
>
> Task|Model|maj@t1|maj@t3|maj@t5|maj@t7|maj@t9|maj@t11
> -|-|-|-|-|-|-|-
> MATH|Ministral|58.2|61.8|63.2|63.6|63.6|63.6
> ||Llama|55.8|57.8|58.4|58.4|58.2|58.4
> GSM8K|Ministral|87.8|89.1|89.1|89.2|89.5|89.2
> ||Llama|87.5|88.2|88.4|88.6|88.6|88.4
>
> Task|Model|pass@t1|pass@t3|pass@t5|pass@t7|pass@t9|pass@t11
> -|-|-|-|-|-|-|-|
> MATH|Ministral|58.2|68.2|70.0|70.8|70.8|70.8
> ||Llama|55.8|61.2|62.0|62.0|62.0|62.4|
> GSM8K|Ministral|87.8|91.6|92.7|93.0|93.3|93.3
> ||Llama|87.5|90.7|91.2|91.5|91.6|91.7
>
> We further conducted analysis on failure patterns in reply to Reviewer 2FHw, Table 1. In qualitative analysis beyond what is reported in the paper, we observed that *iterative refinement can help correct over-refined answers*. For instance, a correct answer may be incorrectly altered due to faulty feedback, but later iterations may recover the correct answer, leading to correct majority voting answer.
>
> While iterative refinement helps mitigate over-refinement, it does not entirely eliminate the risk. As a potential solution, we propose monitoring performance on a validation set at each refinement step and stopping early if accuracy begins to decline.

---

> > ### Comment · Reviewer_jmss · 2025-04-02
> >
> > Thanks for including these additional results with more models and examining the potential effect of over-refinement. These new results largely address my concerns on the results side and I am raising my score accordingly.

---

### Official Review · Reviewer_Hay1 · 2025-03-13

**Overall Recommendation:** 4

**Summary:**

This paper introduces DPSDP (Direct Policy Search by Dynamic Programming), a reinforcement learning algorithm for training multi-agent LLM systems to iteratively refine responses on reasoning tasks. The authors formulate the multi-turn refinement process as a Markov Decision Process with an actor that generates answers and a critic that provides feedback. The algorithm uses direct preference learning on self-generated data to optimize both agents together. A key contribution is the practical adaptation that allows models to generalize to out-of-distribution horizons at test time through a simplified state representation. Theoretical analysis shows that DPSDP achieves performance equivalent to any comparator policy covered in the training distribution.

**Claims And Evidence:**

The paper's claims are generally well-supported by empirical evidence and theoretical analysis. The authors make two principal claims: (1) DPSDP improves reasoning performance through multi-agent interaction; and (2) the approach generalizes to out-of-distribution benchmarks. These claims are substantiated through comprehensive experiments across multiple model families and mathematical reasoning benchmarks.

**Essential References Not Discussed:**

None.

**Experimental Designs Or Analyses:**

The experimental design and analyses in this paper are generally sound. The authors evaluate their approach on appropriate mathematical reasoning benchmarks (MATH 500, GSM8K, MMLU-Pro Math, Olympiad Bench) using standard metrics (pass@turn1, maj@turn5, pass@turn5) that effectively measure both initial and refined performance.

One minor issue is while the authors mention hyperparameter selection in the appendix, a more systematic hyperparameter sensitivity analysis would strengthen the experimental rigor.

**Methods And Evaluation Criteria:**

The methods and evaluation criteria employed in this paper are appropriate and well-suited to the problem of improving LLM reasoning through multi-agent collaboration.

**Other Comments Or Suggestions:**

- line261 h=0, H-1

**Other Strengths And Weaknesses:**

**Strengths**
1. The paper effectively adapts PSDP to the context of LLM-based agent training, creating a theoretically grounded yet practical algorithm for multi-agent response refinement.
2. The authors develop several practical modifications to make the algorithm computationally efficient, particularly the Markovian state reformulation that enables generalization to longer refinement horizons at test time than seen during training.
3. The method shows consistent improvements across different base models and benchmarks.

**Weaknesses**
1. While the experiments cover two model families (Ministral and Llama), they only use 8B parameter versions. Testing with a wider range of model sizes like meta-llama/Llama-3.2-3B-Instruct, meta-llama/Llama-3.3-70B-Instruct would better demonstrate scalability and generalizability of the approach.
2. The evaluation focuses exclusively on mathematical reasoning tasks. Including other reasoning domains (e.g., logical reasoning, coding) would provide a more comprehensive assessment of the method's capabilities.
3. While the paper compares against relevant baselines, it would benefit from comparisons to SCoRe[1] and RISE[2].

[1] Kumar, Aviral, et al. "Training language models to self-correct via reinforcement learning." arXiv preprint arXiv:2409.12917 (2024).
[2] Qu, Yuxiao, et al. "Recursive introspection: Teaching language model agents how to self-improve." Advances in Neural Information Processing Systems 37 (2024): 55249-55285.

**Questions For Authors:**

1. The paper demonstrates results with five answer attempts, but it's unclear how this number was determined. What process did you use to select the optimal number of refinement iterations?
2.  The paper lacks a detailed comparison of computational efficiency between DPSDP and baseline methods like STaR and STaR-DPO. Could you provide quantitative metrics on training time, inference costs?
3. The paper mentions using oracle models (Mistral-Large-Instruct-2411 and Llama-3.3-70B-Instruct) for generating high-quality feedback and refined answers during the preliminary training phase. How essential is this oracle guidance for DPSDP's performance?

**Relation To Broader Scientific Literature:**

This paper builds on and extends several key research directions in the LLM reasoning literature. The verify-and-improve paradigm it explores connects to prior work on self-correction and external feedback mechanisms. The formulation as an MDP extends the application of reinforcement learning approaches to LLM alignment, building particularly on Direct Preference Optimization (Rafailov et al., 2023) and Policy Search by Dynamic Programming (Bagnell et al., 2003).

**Theoretical Claims:**

I reviewed Theorem 1 and its supporting Lemmas (Lemma 2 and Lemma 3), which establish the theoretical performance guarantee for the DPSDP algorithm. The proofs appear to be mathematically sound with appropriate use of Markov Decision Process theory.

One minor issue is that the formal relationship between the practical implementation (Algorithm 2) and the theoretical version (Algorithm 1) could be more explicitly addressed in the theoretical analysis, particularly regarding how the practical approximations affect the theoretical guarantees.

---

> ### Author Rebuttal · Authors · 2025-04-01
>
> Thanks for reviewing our paper!
>
> # Theoretical Claims
> **Q: Analysis of practical algorithm**
>
> We analyze the Q-value approximation in practical DPSDP, where only one feedback and refinement step is used during training (Algorithm 2), assuming $H=3$. Let $\hat{\pi}$ be the resulting policy, and let $\widetilde{Q}_h^{\hat{\pi}}$ denote the estimated Q-values, replacing $Q_h^{\hat{\pi}}$ in Assumption 2.
>
> We define advantage function as $A_h^\pi(s_h,a_h)=Q_h^{\pi}(s_h,a_h)-V_h^{\pi}(s_h)$, and $\widetilde{A}_h^\pi(s_h,a_h)=\widetilde{Q}_h^{\pi}(s_h,a_h)-\mathbb{E}\_{a_h\sim \pi(\cdot\mid s_h)}[\widetilde{Q}_h^{\pi}(s_h,a_h)]$.
>
> As detailed in Section 3.3 (*Estimation of Q-values*), we define the estimated Q-values as follows:
> 1. At $h=2$, the estimated Q is exact.
> 2. At $h=1$, the estimated Q is: $\mathbb{E}\_{a_2 \sim \pi_\mathsf{ref}(\cdot \mid s_2)}[r(s_3)] = Q_1^{\pi_\mathsf{ref}}(s_1, a_1)$.
>
>    We define the approximation error:
>    $$
>    \Delta = \mathbb{E}_{s_h \sim d_h^{\pi^\star},a_h \sim \pi^\star(\cdot \mid s_h)}[A_h^{\hat{\pi}}(s_h,a_h) - \widetilde{A}_h^{\hat{\pi}}(s_h,a_h)]
>    $$
> 3. At $h=0$, we have $\widetilde{Q}_0^{\hat{\pi}_1}(s_0, a_0) = r(s_1) + \frac{H-1}{2} = Q_0^{\pi^\star}(s_0, a_0)$. Therefore,
>
>    $$
>    \mathbb{E}\_{a_h \sim \pi^\star(\cdot \mid s_h)}[A_h^{\hat{\pi}}(s_h, a_h)]
>    \approx \mathbb{E}\_{a_h \sim \pi^\star(\cdot \mid s_h)}[A_h^{\pi^\star}(s_h, a_h)] = 0,
>    $$
>    where the last equality follows from the definition of $A_h^{\pi}$.
>
> Following steps in Appendix C.1, we obtain the approximate upper bound by adding $|\Delta|$ to the theoretical bound.
>
> To assess the impact of $|\Delta|$, we performed an ablation using the step-by-step DPSDP variant (Appendix E.5), which uses $Q^{\pi_2}$ in the DPO-style loss. The results showed no significant performance gain, indicating that $|\Delta|$ has minimal effect. For simplicity and efficiency, we use the original version in the main paper.
>
> # Weaknesses
>
> **Q: Other model sizes and families**
>
> We further tested our algorithm on Qwen2.5-3B to demonstrate its effectiveness over different model sizes and model families.
>
> Task|Model|Pass@t1|Maj@t3|Maj@t5|Pass@t5
> -|-|-|-|-|-
> MATH500|Qwen2.5-3B|57.6|50.0|48.0|58.6
> ||SFT|60.0|60.4|60.4|64.6
> ||DPSDP|60.4|61.6|62.0|65.2
> GSM8K|Qwen2.5-3B|78.6|76.2|75.2|79.4
> ||SFT|79.1|78.4|77.7|81.5
> ||DPSDP|79.9|80.2|79.9|84.2
> MMLU-ProMath|Qwen2.5-3B|47.4|42.0|41.2|48.4
> ||SFT|50.9|51.2|51.4|56.0
> ||DPSDP|52.6|53.2|53.2|57.1
> Olympiad-Bench|Qwen2.5-3B|24.0|22.6|22.0|24.5
> ||SFT|23.9|24.3|24.8|26.4
> ||DPSDP|24.0|23.9|24.0|26.0
>
> Our results show that the proposed algorithm generalizes effectively on smaller models such as Qwen2.5-3B. Furthermore, DPSDP-trained models demonstrate strong generalization capabilities on out-of-distribution benchmarks, such as MMLU Pro Math.
>
> **Q: Other reasoning tasks**
>
> While our focus has been on mathematical reasoning to showcase the effectiveness of our approach, we anticipate that similar performance gains would extend to other complex reasoning tasks. Exploring these tasks presents an exciting direction for future research.
>
> **Q: SCoRe and RISE baselines**
>
> See reply to Reviewer X1o306, Table 1.
>
> # Questions For Authors
>
> **Q: Define the optimal number of refinement iterations**
>
> We scaled the number of refinement iterations up to 10. Our results show that accuracy begins to plateau after approximately 5 to 7 iterations as presented in reply to Reviewer jmss, Table 3.
>
>
> **Q: Comparison on training and inference cost with baselines**
>
> Both DPSDP and baselines have similar time complexity, dominated by quadratic-scaling attention mechanism during forward and backward passes. On 4× H100 80GB GPUs, DPSDP and STaR-DPO each took ~6 hours to train (5h 45m and 5h 55m, respectively), while STaR completed in 3h 10m. Inference costs are comparable, with responses all refined through 4 iterations.
>
> **Q: How essential is the preliminary training (SFT) stage for DPSDP's performance?**
>
> Prior work [1,2] has shown that SFT is essential before reinforcement learning RL, especially when the base model struggles to follow instructions.
>
> Our results support this: we trained DPSDP on Mistral-8B-instruct both with and without SFT. Without SFT, performance degrades notably on  MATH and OlympiadBench, and iterative refinement offers little benefit—as seen in the small gap between accuracy@t1 and majority@t5 or pass@t5.
>
> Task|ModelVariant|Pass@t1|Maj@t5|Pass@t5|
> -|-|-|-|-|
> GSM8K|DPSDP|87.8|89.1|92.7|
> ||w/o SFT stage|90.6|90.8|90.9|
> MATH500|DPSDP|58.2|63.2|70.0|
> ||w/o SFT stage|52.6|53.8|54.4|
> MMLU-ProMath|DPSDP|53.1|54.2|64.3|
> ||w/o SFT stage|54.1|54.4|55.5|
> OlympiadBench|DPSDP|25.8|27.0|32.9|
> ||w/o SFT stage|26.0|26.1|26.7|
>
> These findings highlight the importance of SFT in enabling models to give and use feedback effectively.
>
> [1] Recursive Introspection: Teaching Language Model Agents How to Self-Improve
>
> [2] SFT Memorizes, RL Generalizes: A Comparative Study of Foundation Model Post-training

---

> > ### Comment · Reviewer_Hay1 · 2025-04-03
> >
> > Thank you for your detailed rebuttal and comprehensive experimental results. I have raised my evaluation score.

---

### Decision · Program_Chairs · 2025-05-01

**Decision:**

Accept (poster)

**Comment:**

The authors introduce DPSDP (Direct Policy Search by Dynamic Programming), a novel reinforcement learning algorithm that trains a multi-agent LLM system (actor and critic) to iteratively refine responses for reasoning tasks. They formulate the multi-turn refinement process as a Markov Decision Process and use direct preference learning on self-generated data to jointly optimize the actor and critic. A practical adaptation allows the model to generalize to out-of-distribution refinement horizons at test time. Theoretical analysis suggests DPSDP can achieve performance comparable to any comparator policy within the training distribution. Experiments on mathematical reasoning benchmarks demonstrate that DPSDP improves accuracy with multiple answer attempts. Reviewers highlight the theoretically grounded yet practical algorithm for multi-agent response refinement, the effective adaptation of PSDP to LLM agent training, and the consistent improvements across different models and benchmarks.

Reviewers identify several limitations. Some believe the theoretical analysis is not highly relevant to the practical algorithm due to modifications for feasibility. The baseline comparisons are sometimes limited, with baselines not implemented on all models. The method's intuition could be clearer, particularly regarding pairwise data collection. The paper primarily focuses on mathematical reasoning tasks, limiting the assessment of its capabilities in other domains. While the paper compares against SFT-based baselines, one reviewer suggests a more direct comparison with a fully trained SFT model under controlled training conditions.

For the final version of the paper, please:
- Provide further analysis on how approximations in the practical algorithm affect the theoretical results.
- Include baseline implementations across all evaluated models for a more comprehensive comparison.
- Offer a clearer explanation of the method, especially the pairwise data collection and preference labeling process.
- Further investigate the stopping criterion for the refinement process to mitigate potential over-refinement.
- Analyze failure patterns in more detail to understand the types of errors the method effectively corrects and where it struggles.

Overall though the strengths outweigh the weaknesses and I recommend acceptance. The paper presents a novel and theoretically grounded approach to improving LLM reasoning through a multi-agent reinforcement learning framework. The empirical results demonstrate promising performance gains on challenging mathematical reasoning tasks, showcasing the potential of the method. The algorithm's ability to generalize to out-of-distribution refinement iterations is a significant strength. Furthermore, the reviewers generally acknowledge the sound methodology and the meaningful contribution to the field of LLM reasoning and self-improvement. The authors' efforts to address reviewer concerns in the rebuttal, including providing additional experimental results and clarifications, further strengthen the case for acceptance. It is important to improve the paper either way, incorporating the suggestions from reviewers.